# Hyphal Als proteins act as CR3 ligands to promote immune responses against *Candida albicans*

Tingting Zhou [1], Norma V. Solis[2], Michaela Marshall [3], Qing Yao[4,6], Rachel Garleb[1], Mengli Yang[1,7], Eric Pearlman[3], Scott G. Filler [2,5] & Haoping Liu [1] ✉

Patients with decreased levels of CD18 (β2 integrins) suffer from life-threatening bacterial and fungal infections. CD11b, the α subunit of integrin CR3 (CD11b/CD18, $\alpha_M\beta_2$), is essential for mice to fight against systemic *Candida albicans* infections. Live elongating *C. albicans* activates CR3 in immune cells. However, the hyphal ligands that activate CR3 are not well defined. Here, we discovered that the *C. albicans* Als family proteins are recognized by the I domain of CD11b in macrophages. This recognition synergizes with the β-glucan-bound lectin-like domain to activate CR3, thereby promoting Syk signaling and inflammasome activation. Dectin-2 activation serves as the "outside-in signaling" for CR3 activation at the entry site of incompletely sealed phagosomes, where a thick cuff of F-actin forms to strengthen the local interaction. In vitro, CD18 partially contributes to IL-1β release from dendritic cells induced by purified hyphal Als3. In vivo, Als3 is vital for *C. albicans* clearance in mouse kidneys. These findings uncover a novel family of ligands for the CR3 I domain that promotes fungal clearance.

Invasive candidiasis, caused mainly by the opportunistic fungus *Candida albicans*, is a leading cause of nosocomial bloodstream infection in developed countries, with a mortality that can exceed 40% despite antifungal treatment[1]. The limited efficacy of antifungal drugs and the rising incidence of antifungal resistance indicate a need to develop new therapies for invasive candidiasis[1]. Opportunistic "high-damaging" *C. albicans* strains produce toxic factors, such as pore-forming candidalysin, aspartyl proteases (Saps), and the agglutinin-like sequence (Als) proteins, during the transition from benign commensal yeasts to pathogenic hyphae, exacerbating inflammation[2–5]. Compared to yeasts, hyphae cause more tissue damage, both directly via candidalysin and Saps and indirectly by exacerbating the inflammatory responses. However, the interactions between hyphae and myeloid cells that drive antifungal immunity remain poorly understood.

Cell-surface receptors detect conserved pathogen-associated molecular patterns (PAMPs), activating the host cell's first line of defense known as pattern-triggered immunity (PTI)[6]. Cell surface receptors, such as Dectin-1, Dectin-2, and CR3 that participate in recognizing fungal components rely on Syk for downstream signaling[7–10]. β-glucans and α-mannans are well-characterized *C. albicans* PAMPs[11]. Dectin-1 and Dectin-2, members of C-type lectin receptors (CLRs), recognize *C. albicans* yeast cells and hyphae by binding to surface β-glucans and α-mannans, respectively[12–15]. Mice deficient in Dectin-1 or Dectin-2 have increased susceptibility to systemic candidiasis[14,15]. CR3 also recognizes β-glucans and is vital for the host defense against *C. albicans* infection in the mouse model[16]. Patients with leukocyte adhesion deficiency type I (LAD-I), who express reduced levels of CD18, have impaired neutrophil trafficking into the

[1]Department of Biological Chemistry, University of California, Irvine, CA, USA. [2]Division of Infectious Diseases, Lundquist Institute for Biomedical Innovation at Harbor-UCLA Medical Center, Torrance, CA, USA. [3]Department of Physiology and Biophysics, University of California, Irvine, CA, USA. [4]Division of Biology and Biological Engineering, California Institute of Technology, Pasadena, CA, USA. [5]David Geffen School of Medicine at UCLA, Los Angeles, CA, USA. [6]Present address: Gilead Sciences Inc., Foster City, CA, USA. [7]Present address: Zymo Research Corporation, Irvine, CA, USA. ✉e-mail: h4liu@uci.edu

inflamed tissue and suffer from life-threatening bacterial and fungal infections early in life[17]. CR3 recognizes *C. albicans* hyphae mainly through its I domain, while its lectin-like domain appears to influence CD11b/CD18 binding activity by modulating the function of the I domain[18]. The secreted mannoprotein Pra1, whose production is higher in hyphae and suppressed by fetal bovine serum (FBS)[19], can be recognized by the two β2 integrins, including CR3 and CR4 ($\alpha_X\beta_2$)[20]. The CR3 ligands on hyphae other than Pra1 have not been identified previously.

Inflammasomes are intracellular multiprotein complexes that sense danger signals from damaged cells and pathogens. Mice lacking the NLRP3 inflammasome components, NLRP3, ASC, and caspase-1, are highly susceptible to systemic candidiasis[21–23]. The caspase-1-dependent cytokines, IL-1β and IL-18, are required for the initiation of the protective Th1 and Th17 cellular responses[22,24]. The IL-17AR system is required for normal fungal host defense in vivo[25]. Therefore, the NLRP3 inflammasome activation is crucial for bridging innate and adaptive immunity. The NLRP3 inflammasome can be activated by diverse stimuli and cellular events. β-glucans, the most abundant cell wall component of fungi, are recognized by cell surface receptors, such as Dectin-1 and CR3, and cytoplasmic inflammasome components, resulting in proinflammatory gene transcription and NLRP3 inflammasome activation[26–28]. *C. albicans* hyphae activate the NLRP3 inflammasome significantly more than yeasts, leading to increased production of IL-1β[21,23,29,30]. Candidalysin and Saps also trigger NLRP3 inflammasome activation[31,32]. Here, we demonstrate that hypha-specific Als3 is involved in hypha-mediated NLRP3 inflammasome activation and IL-1β release. The β2 integrins of immune cells, mainly CR3 of macrophages, function as the cell surface sensor of Als proteins. We demonstrated that CR3 physically binds to the hyphal Als3 protein, leading to both Als3-mediated Syk signaling and inflammasome activation.

## Results

### *C. albicans* hyphal Als3 is essential for optimal NLRP3 inflammasome activation

NLRP3 inflammasome activation is a tightly regulated process that requires both priming and activation signals. The priming step induces NLRP3 and IL-1β gene upregulation. The subsequent activation step leads to the assembly of the inflammasome and the cleavage of pro-IL-18 and pro-IL-1β by caspase-1 into their mature forms, allowing their release. Priming macrophages with GM-CSF enhances inflammasome signaling by polarizing anti-inflammatory M2 bone marrow-derived macrophages (BMDMs) toward a glycolytic, anti-microbial M1 phenotype[33,34]. We found that live elongating hyphae induced the primed BMDMs or the J774A.1 murine macrophage cell line to release significantly higher levels of IL-1β compared to yeast, or yeast-locked strain *flo8*[Δ/Δ] or *efg1*[Δ/Δ] (Fig. 1a, b and Supplementary Fig. 1a). Strains defective in hyphal growth, including *brg*[Δ/Δ] and *hgc1*[Δ/Δ], also induced significantly lower levels of IL-1β (Fig. 1b and Supplementary Fig. 1a). Among *C. albicans* mutants with individual deleted genes encoding hyphal surface or secreted proteins, only the *als3*[Δ/Δ] mutant exhibited a defect in both stimulating the release of IL-1β (Fig. 1b) and inducing macrophage death (Fig. 1c and Supplementary Fig. 1b, c).

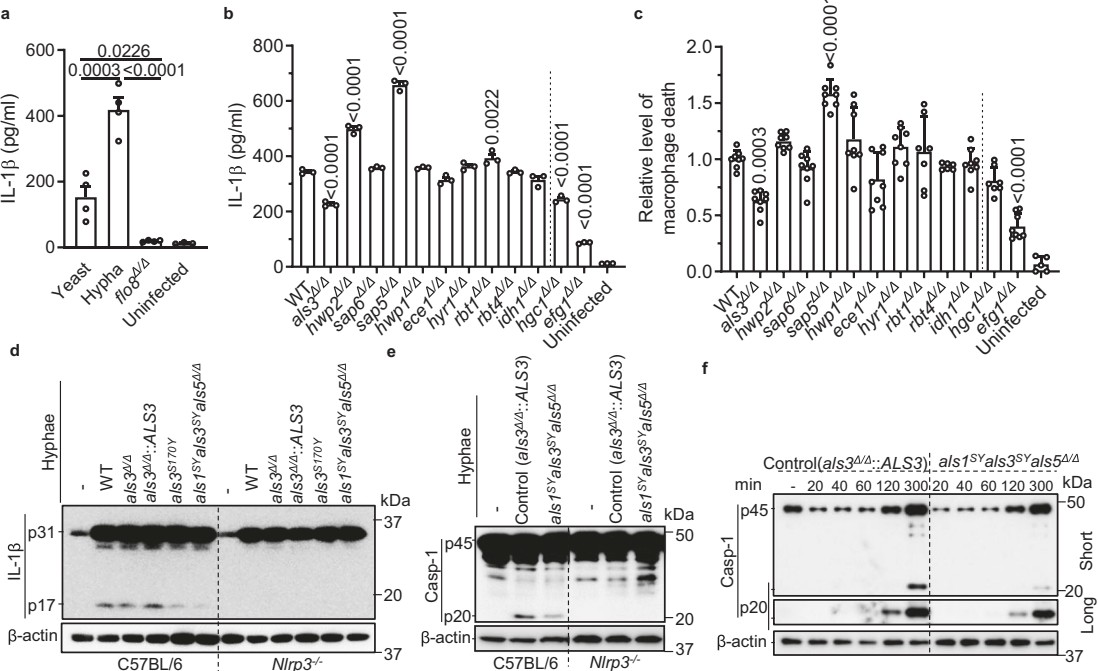

**Fig. 1 | Hyphal Als family proteins promote NLRP3 inflammasome activation.**
**a** *C. albicans* hyphae on plates were used to challenge macrophages at MOI 1. BMDMs were left uninfected or infected with live yeast or hyphal form of *C. albicans* SN250 or *C. albicans* yeast-locked strain *flo8*[Δ/Δ] at MOI 1 for 8 h. Data are pooled from two independent experiments ($n = 4$, infected; $n = 3$, uninfected). **b, c** M1 macrophages were infected with indicated *C. albicans* strains in hyphal form and assessed for IL-1β release and cell death at 6 h post-infection. The levels of IL-1β release were determined by ELISA. Data in **b** are representative of at least three independent experiments ($n = 3$ biological replicates). Cell death was determined by kinetically measuring the uptake of Sytox Green. Data in **c** are pooled from three independent experiments ($n = 5$, uninfected; $n = 6$, *rbt4*[Δ/Δ]; $n = 7$, *rbt1*[Δ/Δ] and *hgc1*[Δ/Δ];

$n = 8$, all others). Dashed lines are used to separate the screening mutants from the control mutants. **d–f** M1 macrophages were infected with the indicated hyphae for 5.5 h. The *als3*[Δ/Δ]::*ALS3* complemented strain showed similar inflammasome activation to WT **d** and served as the control **e, f**. Pro-IL-1β (p31), the cleaved mature IL-1β (p17), procaspase-1 (p45), and the active caspase-1 subunit (p20) in whole cell lysates were analyzed by Western blotting. Data are representative of three independent experiments. *P* values in **a–c** were determined by one-way ANOVA (Tukey's multiple comparisons test) and data in **b, c** were compared with WT. Data in **d** are representative of at least three independent experiments. Data in **a–c** are presented as mean ± SEM. Coincubation was performed with Opti-MEM I medium in **b, c**. Source data are provided as a Source Data file.

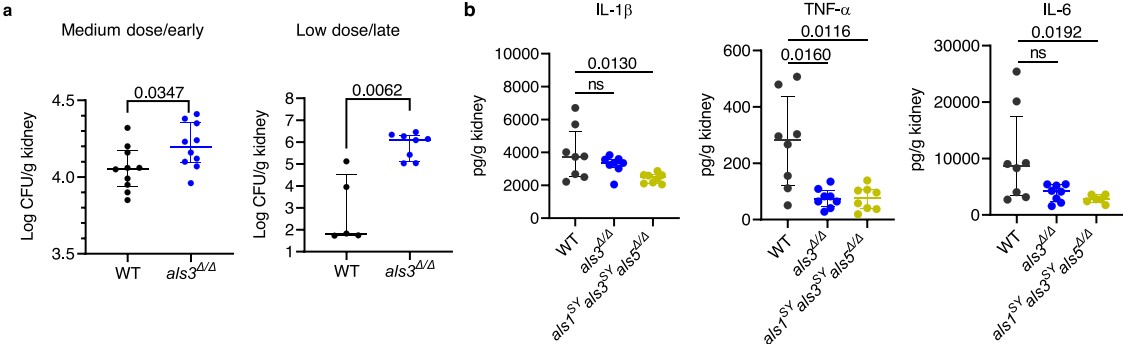

**Fig. 2 | Hyphal Als family protein promotes fungal clearance in kidneys. a** The kidney fungal burden of surviving mice after 18 h (left, 5 males and 5 females) or 19 days of infection (right, 15 female mice infected totally; 5 mice survived after WT infection; 8 mice survived after the $als3^{\Delta/\Delta}$ mutant infection). Medium dose, $1 \times 10^5$; low dose, $7.5 \times 10^4$. $P$ values were determined by two-tailed Mann-Whitney test. **b** Level of indicated cytokines in kidneys after 2 days of infection. 8 female C57BL/6 mice were infected with each strain. $P$ values were determined by Kruskal-Wallis test. ns, not significant. Results were median with interquartile range. Source data are provided as a Source Data file.

Als proteins are a family of GPI-anchored cell wall adhesins consisting of eight large cell surface glycoproteins with high sequence similarity (from Als1 to Als7 and Als9). The three most highly expressed Als proteins are Als3, Als1, and Als5. Among them, Als3 is the most abundant and is also hyphal-specific (16). Als1, Als3, or Als5 has broad substrate specificity and thus mediate adherence to a variety of host constituents[35]. The peptide-binding cavity (PBC) of Als proteins plays an important role in Als3-mediated adhesion[36]. We found that the $als3^{S170Y}$ strain with a mutation of S170 to Y in the PBC was also defective in stimulating the release of mature IL-1β (p17) as compared with wild-type (WT) $C. albicans$ strain SN250 and the complemented $als3^{\Delta/\Delta}::ALS3$ strain (Fig. 1d). The double mutants, $als1^{\Delta/\Delta}als3^{\Delta/\Delta}$, $als1^{\Delta/\Delta}als3^{S170Y}$, and $als3^{S170Y}als5^{\Delta/\Delta}$, didn't show significant difference in inducing cell death compared to the $als3^{\Delta/\Delta}$ mutant (Supplementary Fig. 1c). The $als1^{S170Y}als3^{S170Y}als5^{\Delta/\Delta}$ ($als1^{SY}als3^{SY}als5^{\Delta/\Delta}$ in short) triple mutant induced lower level of IL-1β compared to the $als3^{S170Y}als5^{\Delta/\Delta}$ double mutant (Supplementary Fig. 1d). Immunoblotting results revealed that the $als1^{SY}als3^{SY}als5^{\Delta/\Delta}$ mutant triggered much less active caspase-1 (p20) compared to the complemented control strain $als3^{\Delta/\Delta}::ALS3$ in a NLRP3-dependent manner (Fig. 1e, f). Therefore, Als3 is required for optimal inflammasome signaling during $C. albicans$ infection.

### $C. albicans$ Als proteins, especially Als3, are important for immune activation and fungal clearance

To investigate the role of Als3 in the immune responses against systemic candidiasis, mice were infected with the WT strain and the $als3^{\Delta/\Delta}$ mutant. At 18 h or 19 days post-infection, mice infected with the $als3^{\Delta/\Delta}$ mutant had a significantly higher fungal burden in the kidneys than mice infected with the WT strain (Fig. 2a), indicating that Als3 is essential for fungal clearance from this organ. By contrast, the brain fungal burdens in mice infected with these strains were low and not significantly different (Supplementary Fig. 2a). Therefore, Als3 is required for $C. albicans$ clearance from the kidneys but not the brain.

Given the vital role of cytokines and chemokines in immune responses, we measured the cytokine and chemokine profiles of the kidneys during systemic $C. albicans$ infection. In comparison with mice infected with WT $C. albicans$, those infected with the $als3^{\Delta/\Delta}$ mutant showed a trend towards reduced cytokines and chemokines, including IL-1β, TNF-α, IL-6, IL-10, CXCL1/KC, and CXCL2/MIP-2α, with only the reduction of TNF-α being statistically significant (Fig. 2b and Supplementary Fig. 2b). In contrast, the $als1^{SY}als3^{SY}als5^{\Delta/\Delta}$ strain induced significantly lower levels of all these cytokines and chemokines. Thus, the Als family is required for optimal immune responses.

### N-acetylglucosamine-induced hyphae exhibit higher $ALS3$ expression and inflammasome activation

Hypha-specific genes, such as $HWP1$, $ECE1$, $ALS1$, $ALS3$, $HYR1$, and $SOD5$, are highly upregulated during systemic candidiasis[37]. Hyphal characteristics in vitro vary in different media. N-acetylglucosamine (GlcNAc) is one of the stimuli that induces the expression of hyphal-specific genes and stimulates the yeast-to-hypha transition[38–40]. Quantitative RT-PCR data confirmed that the hyphal expression of $ALS3$, but not $ALS1$ and $ALS5$, was consistently and significantly higher in media containing GlcNAc than media without GlcNAc. The expression pattern was not affected by different basal media, including yeast extract peptone (YEP), synthetic complete (SC), or RPMI 1640 (Fig. 3a and Supplementary Fig. 3a, b).

When BMDMs were infected with hyphae prepared in medium containing GlcNAc, the control strain induced significantly higher levels of processed caspase-1 and mature IL-1β compared to the $als1^{SY}als3^{SY}als5^{\Delta/\Delta}$ mutant (Fig. 3b, c, $P < 0.0001$), while hyphae cultured in the medium without GlcNAc induced a weaker response with the same pattern (Fig. 3b, c, $P < 0.01$). Hyphae of the $als1^{SY}als3^{SY}als5^{\Delta/\Delta}$ mutant had either the same or slightly increased β-glucan exposure (Supplementary Fig. 2b). Thus, the reduced inflammasome activation by the $als1^{SY}als3^{SY}als5^{\Delta/\Delta}$ mutant was not due to lower β-glucan exposure. In addition, caspofungin-induced hyphae served as a negative control (Fig. 3b). The defect of inflammasome activation is likely resulting from impaired hyphal gene expression[41] and β-(1,3)-glucan synthesis by targeting $FKS$ gene products[42–44]. Together, our data demonstrate that N-acetylglucosamine preconditions hyphae for Als-3 mediated inflammasome signaling.

### CR3 is involved in Als-mediated inflammasome activation

The activation of CR3 for complement-dependent cytotoxicity requires its dual ligation with both iC3b and cell wall β-glucan[45]. To investigate whether the dual ligation of Als proteins and cell wall β-glucan activates CR3, WT, and $CD18^{-/-}$ BMDMs were assessed. We observed that infection of $CD18^{-/-}$ BMDMs with the control strain induced lower levels of the cleaved form of caspase-1, IL-1β, and N-terminal Gasdermin D (N-GSDMD) compared to infection of WT BMDMs (Fig. 4a). However, the amount of fungal-induced cell death was independent of CD18 (Supplementary Fig. 4a), consistent with the result from live yeast-infected WT and $CD18^{-/-}$ BMDCs[28]. These results suggest that $CD18^{-/-}$ BMDMs died via an N-GSDMD pore-independent pathway. In contrast, the difference in inducing inflammasome activation between the control strain and the $als1^{SY}als3^{SY}als5^{\Delta/\Delta}$ mutant was lower in $CD18^{-/-}$ BMDMs compared to WT BMDMs (Fig. 4a), suggesting the involvement of CD18 in Als-mediated inflammasome activation. To investigate whether CD18

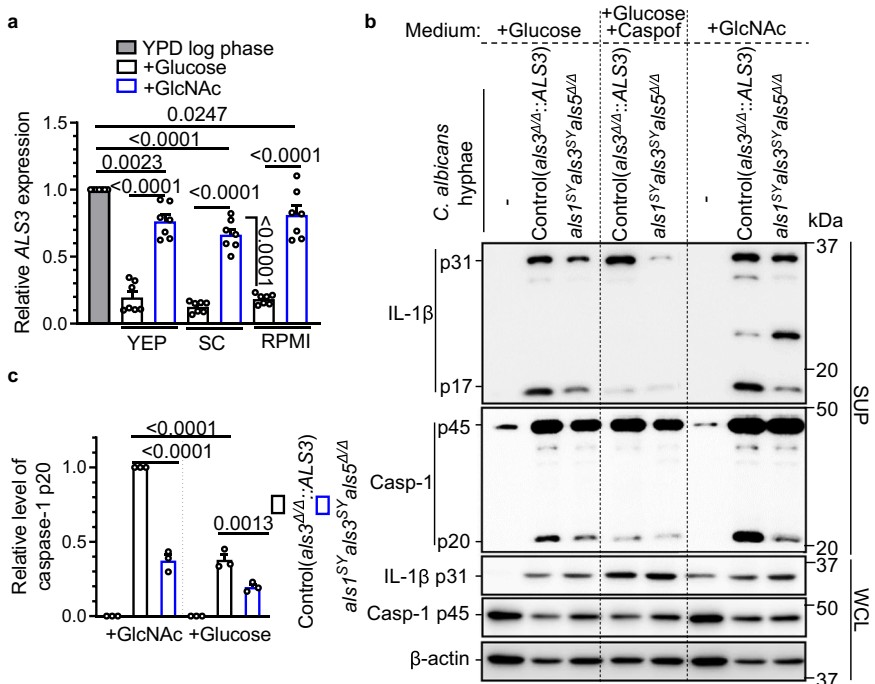

**Fig. 3 | Hyphal Als3 level correlates with the extent of NLRP3 inflammasome activation. a** Quantitative real-time RT-PCR was performed to measure *ALS3* mRNA levels in SC5314 hyphae induced in the indicated medium. Expression levels were normalized to *ACT1* and set to 1 for log-phase SC5314 grown in YPD at 37 °C. Glucose or GlcNAc was included as a carbon source. The values are presented as a scatterplot from two independent experiments out of three (*n* = 7). **b** Immunoblot analysis of caspase-1 and IL-1β from BMDMs at 5.5 h after the infection of the *als3^Δ/Δ^::ALS3* strain (control) or the *als1^SY^als3^SY^als5^Δ/Δ^* mutant. RPMI with glucose,

RPMI with glucose plus caspofungin (caspof, 5 ng/mL), and SC plus GlcNAc were used to induce hyphae. The data are representative of 3 independent experiments. **c** Relative levels of caspase-1 p20 after infection with the indicated hyphae as in **b**. Band intensity of cleaved caspase-1 was quantified by ImageJ software from 3 independent Western blots. Data in **a**, **c** are presented as mean ± SEM. *P* values were determined by one-way ANOVA (Tukey's multiple comparisons test). ns, not significant. Source data are provided as a Source Data file.

collaborated with CD11b in fulfilling this role, we used a CD11b blocking antibody and its isotype control. The CD11b-blocking antibody reduced the released IL-1β level from BMDMs infected with different hyphae by different extents (WT and the *als3^Δ/Δ^::ALS3* strain, *P* < 0.0001; *als* mutant strains, *P* < 0.01 or *P* < 0.05) (Fig. 4b), indicating CD11b is involved in Als-mediated IL-1β release. To exclude the possibility that the results were due to differences in hyphal phagocytosis by the BMDMs of the different mouse strains, we also challenged bone marrow-derived dendritic cells (BMDCs) from C57BL/6 or *CD18^-/-^* mice with purified Als proteins. The deficiency of *CD18^-/-^* significantly dampened Als3-mediated IL-1β release relative to the reference triggers, LPS or curdlan (Fig. 4c and Supplementary Fig. 4b). Also, purified Als3^S170Y^, with a mutated PBC, failed to induced IL-1β production by BMDCs from both strains. These data further indicate that Als3 partially acts through CD18 to mediate immune responses.

Syk kinase signaling is critical for the host defense against fungal pathogens[23]. Immunoblotting of WT M1 BMDMs after *C. albicans* infection showed that both the control strain and the *als1^SY^als3^SY^als5^Δ/Δ^* mutant induced robust Syk phosphorylation that persisted for 1 hour (Fig. 4d). After 2 hours, Syk phosphorylation induced by the control strain returned to basal levels, but it remained high in BMDMs infected with the *als1^SY^als3^SY^als5^Δ/Δ^* mutant (Fig. 4d). CD18 deficiency, but not Dectin-1 deficiency, abrogated this distinction (Fig. 4d), indicating that CR3 is involved in sensing hyphal Als proteins. In addition, the control strain and the *als1^SY^als3^SY^als5^Δ/Δ^* mutant induced comparable phosphorylation of the p38 MAP kinase and similar levels of the inflammasome-independent cytokine TNF-α secretion (Fig. 4d, e), demonstrating that Als proteins do not affect the priming step of

inflammasome signaling in M1 macrophages that have already been primed with GM-CSF. Thus, within M1 macrophages, only inflammasome-dependent cytokine IL-1β was significantly affected by Als proteins (Fig. 4a–c), suggesting that Als proteins directly affect the activation step of inflammasome signaling.

Syk signaling is terminated when the ligated receptor is internalized[46]. We therefore investigated whether the early termination of Syk phosphorylation resulted from the rapid engulfment of hyphae. Phagocytosis of hyphae by macrophages was examined by staining the extracellular hyphae with ConA first and then staining the entire hyphae with calcofluor white after permeabilization. We found that almost all control hyphae were engulfed, while portions of the *als1^SY^als3^SY^als5^Δ/Δ^* mutant hyphae remained on the outside of macrophages at 2 h post-infection. Thus, the control hyphae were phagocytosed more avidly compared to the *als1^SY^als3^SY^als5^Δ/Δ^* hyphae (Supplementary Fig. 4c). These data indicate that Als proteins contribute to macrophage phagocytosis of *C. albicans* hyphae, aligning with the established role of CR3 in mediating phagocytosis[47].

**Hyphal Als proteins interact with the CR3 I-domain, which synergizes with the β-glucan-ligated lectin-like domain, to activate CR3 in macrophages**

To delve deeper into the Als-CR3 interaction, we first examined the binding of macrophage CD18 with the hyphal form of SN250, the *als3^Δ/Δ^* mutant, and the *als3^Δ/Δ^::ALS3* strain after DSP crosslinking. The *als3^Δ/Δ^* mutant exhibited reduced CD18 binding compared to both the WT and the *als3^Δ/Δ^::ALS3* strain (Fig. 5a). In immortalized bone marrow-derived macrophage (iBMDM) lysates with Als3-His or Als3^S170Y^-His, we

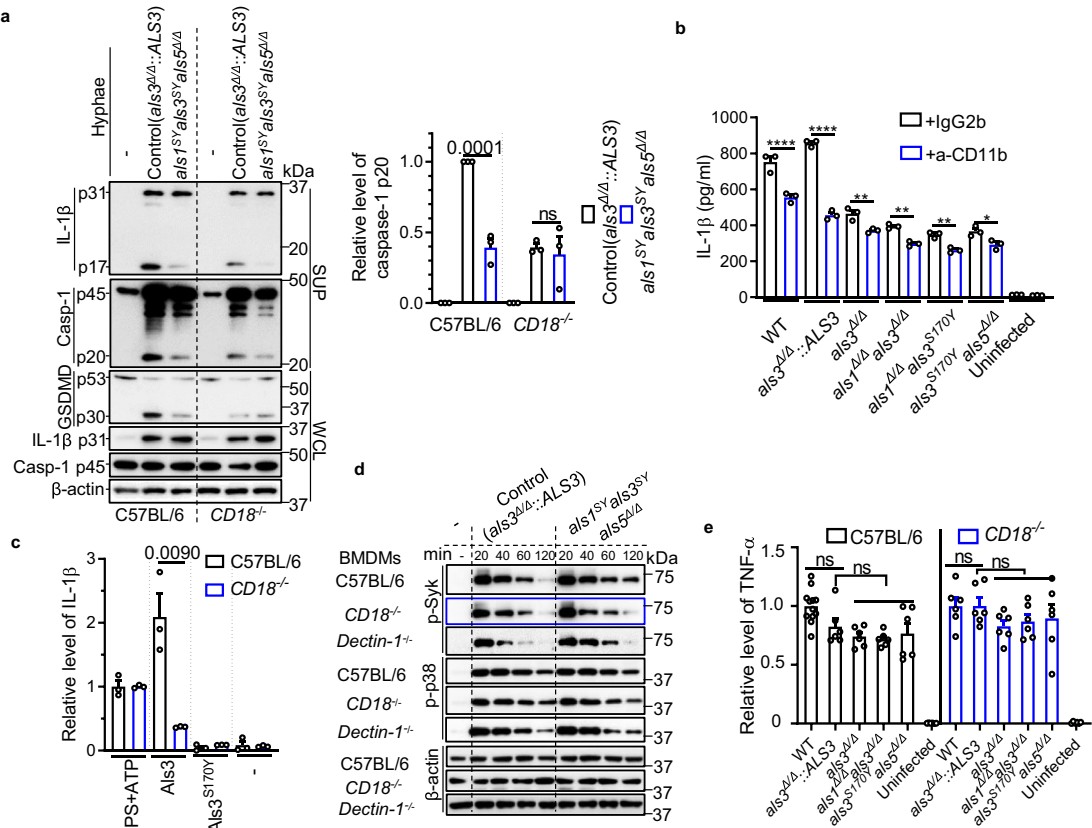

**Fig. 4 | CR3 is involved in Als-mediated inflammasome activation in macrophages. a** Immunoblot analysis of caspase-1, IL-1β, pro- (p53) and activated (p30) GSDMD from C57BL/6 or $CD18^{-/-}$ BMDMs at 5.5 h post-infection with indicated *C. albicans* hyphae. *n* = 3 biologically independent samples. **b** C57BL/6 BMDMs were pretreated with anti-CD11b blocking antibody (M1/70, 10 μg/mL) or control antibody for 30 min before stimulation with indicated *C. albicans* hyphae. IgG2b, κ was used as an isotype control. Macrophages were stimulated with SN250 or the indicated mutants for 5.5 h. The levels of IL-1β release were determined by ELISA. **c** His-tagged hyphal Als3 or Als3$^{S170Y}$ protein (15 μg ml⁻¹) was used to stimulate BMDCs for 24 h. IL-1β concentrations in the culture supernatants were measured by ELISA. Values were normalized to the average value of ATP-stimulated BMDCs (5 mM ATP for 1 h; pre-stimulated with 100 ng/mL LPS for 2 h). **d** Immunoblot analysis of phosphorylated Syk (p-Syk) and phosphorylated p38 (p-p38) from the lysate of BMDMs after infection with the hyphal form of the $als3^{Δ/Δ}::ALS3$ strain (control) or the $als1^{SY}als3^{SY}als5^{Δ/Δ}$ mutant. Data are representative of three independent experiments. **e** BMDMs were infected with the indicated hyphal form of *C. albicans* strains for 5.5 h. TNF-α levels were determined by ELISA. Values were normalized to the average value of SN250-infected samples. Data are pooled from two independent experiments (*n* = 12, WT; *n* = 6, all others). Data in **a**, **b**, **c**, **e** are presented as mean ± SEM; *n* = 3 (**b**, **c** from one representative experiment of three independent experiments). The data in **c** were analyzed by two-tailed *t*-test. *P* values in **a**, **b**, **e** were determined by one-way ANOVA (Tukey's multiple comparisons test). ns, not significant. Hyphae were induced in SC medium with GlcNAc. Source data are provided as a Source Data file.

observed stronger interaction of Als3 with CD11b in comparison with Als3$^{S170Y}$ by immunoprecipitation with CD11b antibodies (Supplementary Figs. 5a, b). These findings strongly suggest that Als3 interacts with CD18 during ingestion of *C. albicans* hyphae by macrophages.

His-tagged Als3 and recombinant CR3, including the CD11b and CD18 subunits, were used to analyze protein-protein interactions. Pulldown assays and immunoblotting of the His tag validated a specific interaction between purified Als3 and recombinant CD11b/CD18, whereas Als3$^{S170Y}$ did not exhibit this interaction (Fig. 5b). The α integrin I-domain possesses certain features, including binding to many extracellular matrix proteins such as fibrinogen and fibronectin[48]. It also contains a metal-ion-dependent adhesion site (MIDAS) that requires divalent cations (such as Mg²⁺ and Mn²⁺) to enhance binding affinity with its ligands[49]. The pull-down results also showed that the binding of Als3 to CR3 is dependent on divalent cation Mn²⁺, and this interaction could be disrupted by fibronectin (Fig. 5b). These data indicate that Als3 associates with the I-domain of CD11b.

To determine whether divalent cations are required for Als-mediated inflammasome activation, we treated the cells with additional magnesium (MgCl₂) or EDTA. Treatment with high MgCl₂ led to an increase in the level of caspase-1 p20 in BMDMs infected with the control strain but not the $als1^{SY}als3^{SY}als5^{Δ/Δ}$ mutant (Fig. 5c). Moreover, EDTA treatment diminished the priming signaling induced by hyphae, as evidenced by the low levels of pro-IL-1β and caspase-1 p20 induced by Als proteins (Fig. 5c). Hence, Als-mediated inflammasome activation is divalent cation-dependent, consistent with our finding of Als3 binding to the I domain of CD11b.

In addition to the I-domain that typically binds extracellular matrix proteins, the lectin-like domain of CR3 binds carbohydrate ligands, such as fungal β-glucan, α- or β-methylmannoside, and α- or β-methylglucoside[50]. Based on this, we performed a pull-down with particulate β-glucan as the solid phase and used soluble β-glucan or β-methylglucoside as antagonists. Als3 greatly increased CR3 binding to particulate β-glucan (Fig. 5d). Soluble β-glucan and β-methylglucoside both exhibited a partial inhibition of the Als3-mediated interaction between CR3 and particulate β-glucan (Fig. 5d). In BMDMs, we also found that hyphal Als-mediated inflammasome activation was largely inhibited by β-methylglucoside and soluble β-glucan (Fig. 5e), indicating that hyphal β-glucan is involved in the Als-mediated inflammasome signaling.

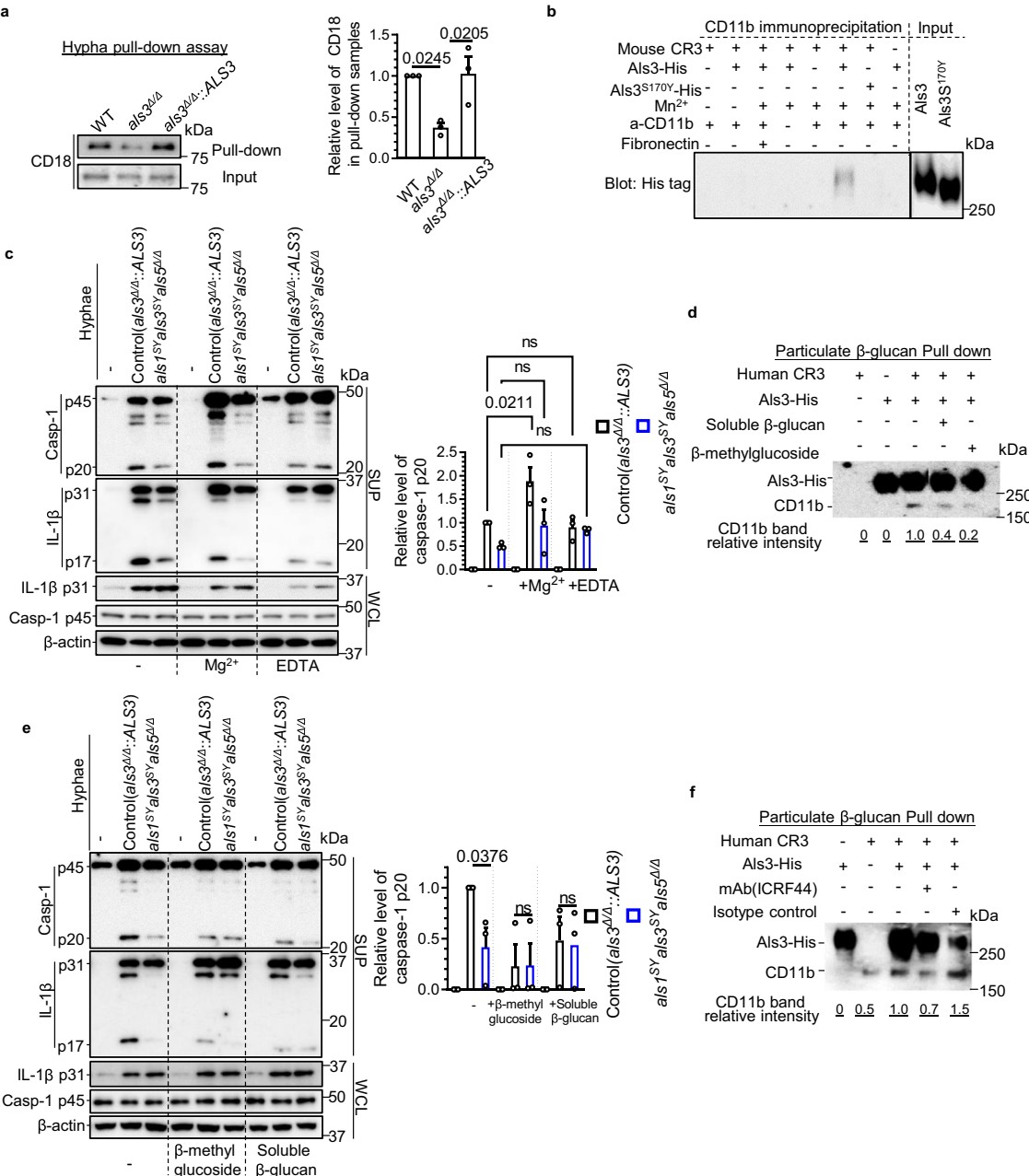

**Fig. 5 | Hyphal Als proteins synergize with cell wall β-glucan to activate CR3 via binding with the I-domain and lectin-like domain respectively. a** Hypha-bound CD18 after the infection of indicated strains was detected by immunoblotting. The whole lysates were used as controls. **b** Purified Als3-His directly interacts with recombinant CR3. Immunoprecipitation of Als3-His or Als3$^{S170Y}$-His with an anti-CD11b antibody in the presence or absence of the indicated factors. **c** Immunoblot analysis of caspase-1 and IL-1β from BMDMs after infection with indicated *C. albicans* hyphae with or without Mg$^{2+}$ (additional 1 mM) or EDTA (5 mM). **d** Particulate β-glucan pull-down assay to analyze the interaction of β-glucan, human CR3, and Als3. Soluble β-glucan or β-methylglucoside was used for competition. **e** Immunoblot analysis of caspase−1 and IL−1β from BMDMs after infection with

indicated *C. albicans* hyphae with or without β-methylglucoside or soluble β-glucan. **f** Particulate β-glucan pull-down assay to analyze the interaction of β-glucan, human CR3, and Als3. Monoclonal antibody ICRF44 (specific for the I-domain) or an isotype control was used for competition. Hyphae were induced in SC medium with GlcNAc. Data in **b**, **d**, **f** are representative of at least three independent experiments. Data in **a**, **c**, **e** are presented as mean ± SEM. *P* values in **a**, **c** were determined by one-way ANOVA (Tukey's multiple comparisons test) (*n* = 3 independent experiments). *P* values in **e** were analyzed by two-tailed t-test (*n* = 3 independent experiments). ns, not significant. Source data are provided as a Source Data file.

In the particulate β-glucan pull-down assay, the Als-mediated interaction of CR3 and β-glucan was also significantly inhibited by the I-domain specific antibody ICRF44, but not its isotype control (Fig. 5f), further supporting an association of Als3 with the CD11b I domain and the synergy between the I domain and the lectin-like domain in ligand binding.

## Als-mediated CR3 clustering at the F-actin cuff promotes Syk signaling and inflammasome signaling

To determine whether the Als-CR3 interaction affects priming, unpolarized macrophages (M0) were used to analyze hypha-induced Syk signaling. In comparison to WT macrophages, *CD18*$^{-/-}$ macrophages exhibited a reduced level of Syk phosphorylation, while the

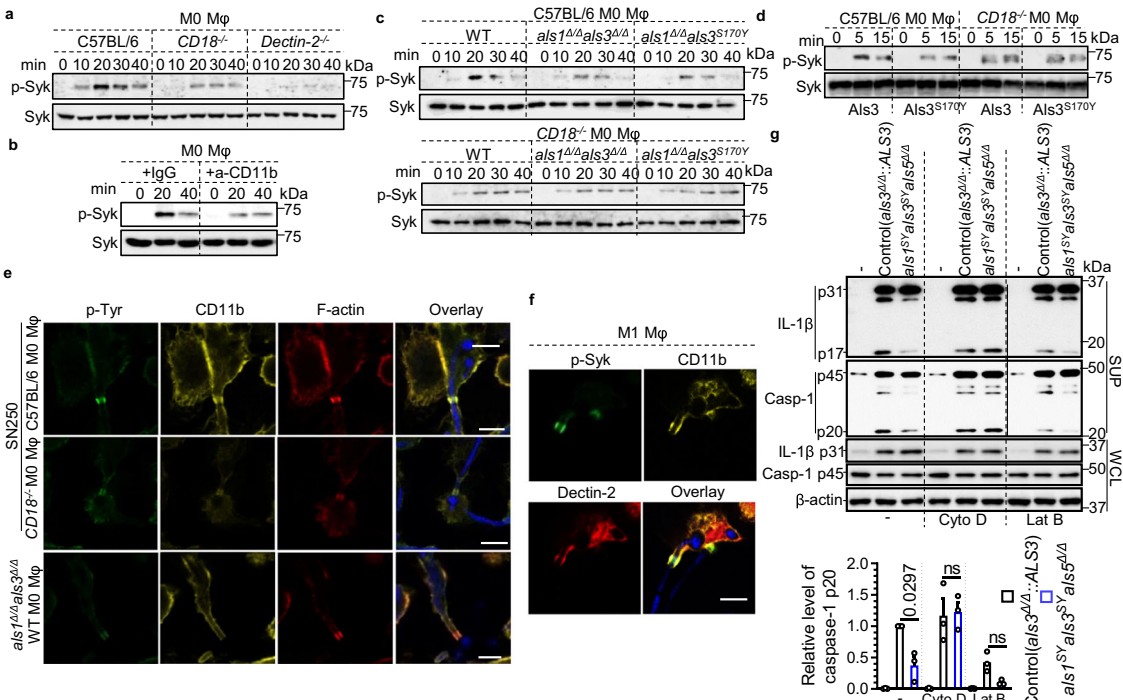

**Fig. 6 | The CR3-Als3 complex at the F-actin cuff promotes Syk phosphorylation and inflammasome activation. a** Immunoblot analysis of p-Syk from C57BL/6 and *CD18*[-/-] M0 BMDM (without GM-CSF priming) lysates after stimulation with *C. albicans* hyphae for indicated periods. Total Syk served as a loading control. **b** WT M0 BMDMs were pretreated with anti-CD11b blocking antibody (M1/70, 10 μg/mL) for 30 min before stimulation with *C. albicans* hyphae. Cell lysates were subjected to immunoblot analysis of p-Syk and total Syk. IgG2b, κ was used as an isotype control. **c** Immunoblot analysis of p-Syk and total Syk from C57BL/6 or *CD18*[-/-] M0 BMDMs after WT, *als1*[Δ/Δ]*als3*[Δ/Δ], or *als1*[Δ/Δ]*als3*[S170Y] hyphal infection. **d** Immunoblot analysis of p-Syk from C57BL/6 and *CD18*[-/-] M0 BMDM (without GM-CSF priming) lysates after stimulation with 15 μg ml[-1] Als3 or Als3[S170Y] for indicated periods. Total

Syk served as a loading control. **e** Immunofluorescence images of M0 BMDMs after stimulation with *C. albicans* hyphae expressing TagBFP for 20 min. p-Tyr, CD11b, and F-actin were analyzed. Scale bars, 7.5 μm. **f** Immunofluorescence images of M1 BMDMs after infection with *C. albicans* hyphae expressing TagBFP for 20 min. p-Syk, CD11b, and Dectin-2 were analyzed. Scale bar, 7.5 μm. **g** Immunoblot analysis of caspase-1 and IL-1β after infection with *C. albicans* hyphae in BMDMs with or without Cyto D and Lat B. Hyphae were induced in RMPI medium **a**–**c**, **e**, **f** and SC medium plus GlcNAc **g**. Each experiment was repeated three times independently. Data in **g** are presented as mean ± SEM. *P* values in **g** were determined by one-way ANOVA (Tukey's multiple comparisons test). ns not significant. Source data are provided as a Source Data file.

absence of Dectin-2 almost completely abolished hypha-induced Syk phosphorylation (Fig. 6a and Supplementary Fig. 6a). Complete inhibition of Syk phosphorylation was observed upon treating macrophages with EDTA (Supplementary Fig. 6b), consistent with the requirement of divalent cations for the activation of both CD18 and Dectin-2[51]. In line with the results from *CD18*[-/-] macrophages, treatment with a CD11b blocking antibody resulted in reduced Syk phosphorylation following hyphal infection compared to the IgG control (Fig. 6b). Therefore, in addition to Dectin-2, CR3 also contributes to Syk signaling during hyphal infection.

In WT macrophages, infection with WT hyphae led to an increase in p-Syk levels at 20 min post-infection, whereas infection with the *als1*[Δ/Δ]*als3*[Δ/Δ] or *als1*[Δ/Δ]*als3*[S170Y] strains resulted in less Syk activation (Fig. 6c and Supplementary Fig. 6c). This difference was not evident when *CD18*[-/-] macrophages were infected (Fig. 6c and Supplementary Fig. 6c), indicating that the increased Syk phosphorylation facilitated by Als originates from the activation of CR3.

Purified Als3 and Als3[S170Y] proteins were used to examine their ability to induce Syk phosphorylation in macrophages. Als3 induced a higher level of Syk phosphorylation compared to Als3[S170Y] in WT BMDMs, whereas this difference was not seen in *CD18*[-/-] macrophages (Fig. 6d). In *Dectin-2*[-/-] BMDMs, Als3- or Als3[S170Y]-induced Syk phosphorylation was below detectable levels (Supplementary Fig. 6d), suggesting that α-mannans of Als3 or Als3[S170Y] contributes to Dectin-2-mediated signaling.

Ligation of fungal β-(1,3)-glucan by CR3 plays a crucial role in the formation of the F-actin cuff during the frustrated phagocytosis of

elongated hyphae[44,52]. We employed immunostaining to examine tyrosine phosphorylation (p-Tyr), CR3, and F-actin localization patterns following hyphal stimulation. As reported[44], both p-Tyr and CD11b were concentrated at the F-actin cuff within BMDMs after *C. albicans* hyphal infection (Fig. 6e). In WT macrophages infected with the *als1*[Δ/Δ]*als3*[Δ/Δ] mutant and *CD18*[-/-] macrophages infected with the WT strain, there was a relatively weak accumulation of p-Tyr at the actin cuff after hyphal stimulation (Fig. 6e and Supplementary Fig. 6e), consistent with the immunoblotting data in Fig. 6c and further demonstrating a requirement for CR3.

Integrins typically exhibit weak binding to their ligands unless cells encounter inflammatory stimuli that trigger "inside-out signaling". This process enhances the avidity and affinity of cell surface integrins for their ligands. When ligands bind to the extracellular domain of integrins, it initiates "outside-in signaling", leading to integrin clustering, recruitment of actin filaments, and signaling proteins to the cytoplasmic domain of integrins[53]. In Dectin-1-expressing RAW264.7 cells, Dectin-1 initiates "inside-out signaling" of CR3 in response to *C. albicans* hyphae[44]. In BMDMs, Dectin-2 co-localizes with CD11b and p-Syk, accumulating at the F-actin cuff (Fig. 6f), indicating that Dectin-2 and CR3 act together to promote Syk signaling. To explore the contribution of Dectin-2 to CR3 activation, immunoprecipitation was performed with the monoclonal antibody (mAb) CBRM1/5 that recognizes an activation-specific epitope of CD11b[54,55]. CD11b was pulled down from WT macrophages, but not from *CD18*[-/-] or *Dectin-2*[-/-] BMDMs after *C. albicans* hyphal infection (Supplementary Fig. 6f). Therefore, we propose that Dectin-2 provides "inside-out

signaling" for CR3 activation, thereby enhancing the binding of Als3 to the I domain of CR3.

Integrin clustering after its full activation rearranges the cytoskeleton, and excludes phosphatases, such as CD45, from the clustering region, and resulting in enhanced phosphorylation-based cellular signaling[56]. To test whether F-actin remodeling is involved in Als-mediated inflammasome activation, Cytochalasin D (Cyto D) and Latrunculin B (Lat B) were used to inhibit F-actin polymerization. Cyto D inhibits the addition of new actin monomers by occupying the filament's barbed end, whereas Latrunculin A/B (Lat A/B) interacts with monomeric actin thereby preventing its self-assembly into filaments. Our results showed that Cyto D treatment weakened the difference between the control strain and the *als1^{SY}als3^{SY}als5^{Δ/Δ}* mutant but did not reduce the level in activating inflammasome. Lat B treatment reduced the levels of inflammasome signaling in both the control strain and the *als1^{SY}als3^{SY}als5^{Δ/Δ}* (Fig. 6g). Thus, F-actin remodeling is essential for Als-mediated inflammasome signaling.

## Discussion

In summary, frustrated phagocytosis of hyphae induces a high concentration of actin at the entry site, wrapping the hyphae, accompanied by a high level of receptors, including CR3 and Dectin-2, and tyrosine kinase signaling. The Als-I domain interaction, synergizing with the cell wall β-glucan-lectin-like domain, acts as "inside-out signaling" for CR3 activation. Dectin-2 activation serves as the "outside-in signaling" for CR3 activation. The resulting activation of CR3 promotes proinflammatory and inflammasome signaling during hyphal infection (Fig. 7).

Earlier studies often used the yeast form of *C. albicans* to infect LPS-primed macrophages at an MOI higher than 5[57]. The first wave of inflammasome activation from receptor activation was only seen with a high MOI (3-6) when a large number of yeasts infected unpolarized macrophages[57,58]. Here, we used GM-CSF-primed macrophages, known as M1 inflammatory macrophages, to study hypha-macrophage interactions at MOI 1. The robust inflammatory property of M1 macrophages enables us to evaluate hyphal cell surface components essential for activating receptors that mediate inflammasome activation.

Als3 is important for *C. albicans* adhesion to endothelial and epithelial cells[36]. *C. albicans* Als3 adheres to endothelial/epithelial cells via E-cadherin/N-cadherin, EGFR, and HER2[59]. CR3 is a surface receptor on a subset of myeloid cells, certain activated lymphoid cells, and some epithelial cells such as cervical[60] and alveolar epithelial cells[61]. This integrin is involved in multiple cellular functions such as phagocytosis, cellular activation, cell-mediated killing, and chemotaxis[62]. Therefore, the Als-CR3 axis is almost certainly involved in multiple important processes. The biological role of CR3 is known in greater detail than that of CR4, and the structural similarity between CR3 (CD11b/CD18) and CR4 (CD11c/CD18) suggests that both receptors are responsible for the phagocytosis of iC3b-opsonized antigens and apoptotic cells[63]. Since we used *CD18^{-/-}* cells in most experiments, we cannot exclude the possibility of the interaction between Als proteins and CR4.

Cell surface receptors, such as CLRs and CR3, activate Syk signaling to promote proinflammatory immune responses. Dectin-2 and Dectin-3 recognize hyphal α-mannan and promote *C. albicans* clearance[15,64]. CD11b also recognizes *C. albicans* hyphae[18] and plays a critical role in host defense against *C. albicans* infection. Our study showed that CR3 contributes to Als3-mediated inflammasome activation, in agreement with the previous report that CR3, but not Dectin-1, is partially required for live *C. albicans* yeast-induced IL-1β release[28].

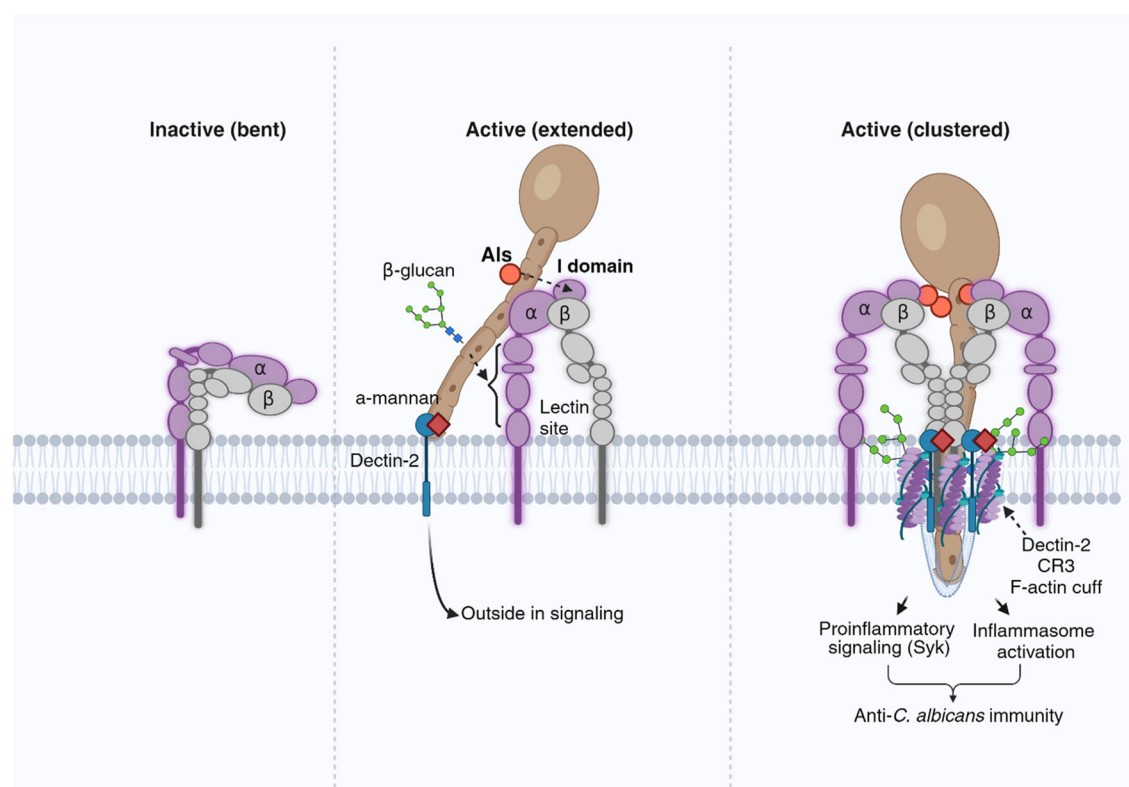

**Fig. 7 | The Als-CR3 axis promotes proinflammatory signaling and inflammasome activation.** Hyphal Als proteins are recognized by the host CR3 via its I domain. The Als-bound I domain synergizes with the β-glucan-ligated lectin-like domain, acting as the "inside-out signaling" for CR3 activation. CR3 and Dectin-2 are enriched at the F-actin cuff, where tyrosine kinase signaling is concentrated and Dectin-2 activation serves as the "outside-in signaling" for CR3 activation. Consequently, Als proteins, as new ligands of the CD11b I domain, promote proinflammatory and inflammasome signaling. Adapted from "Outside-in and Inside-out Integrin Signaling Pathways", by BioRender.com (2024). Retrieved from https://app.biorender.com/biorender-templates.

We found that CR3 controls Als-mediated Syk signaling. Although Syk is essential for inflammasome activation by live *C. albicans* yeast[23], we showed that CR3 deficiency did not completely abolish Als-mediated inflammasome activation, suggesting other receptors participate in Als3-mediated inflammasome activation.

The interactions among PRRs play a crucial role in the innate immune response against microbes[65]. CR3 collaborates with multiple receptors, such as TLR2 and Dectin-1, contributing to the elimination of microbes from the host[28,55,66]. During the frustrated phagocytosis, there is a massive reorganization of membrane proteins, and membrane movement coordinates with the actin cytoskeleton to trap hyphae. In line with this, we showed that CR3 directs Als3-mediated F-actin remodeling during hyphal infection. Our results also demonstrated that the co-activation of Dectin-2 and CR3 by hyphae occurred at the contact site where tyrosine kinase phosphorylation and F-actin cuff were detected, emphasizing the importance of initial contact in triggering proinflammatory responses.

In conclusion, this study identified hyphal Als proteins as new ligands of the CR3 I domain, which synergizes with β-glucan to activate CR3, promoting Syk signaling and inflammasome signaling during infection. Als3-mediated fungal clearance is essential for preventing *C. albicans* overgrowth.

# Methods

## Ethics
All studies described have been approved by UC Irvine oversight committees, including the use of mice as a source of macrophages and dendritic cells by the UC Irvine IACUC. The mouse systemic infection studies were approved by the Institutional Animal Care and Use Committee at the Lundquist Institute for Biomedical Innovation at Harbor-UCLA Medical Center.

## Mice
$CD18^{-/-}$ mice were originally provided by Claire Doerschuk (University of North Carolina, Chapel Hill, NC, USA). Dectin-2-deficient mice were a kind gift from Yoichiro Iwakura (University of Tokyo, Tokyo, Japan). Age- and sex-matched C57BL/6 J mice were purchased from Jackson Laboratories (Bar Harbor, ME, USA). 6-week-old male and female C57BL/6 J mice were purchased for in vivo experiments. All animals were bred in pathogen-free conditions in microisolator cages and were treated according to institutional guidelines following approval by the University of California IACUC.

## Culture of *C. albicans*
Single colonies of *C. albicans* strains from yeast-peptone-dextrose (YPD) solid medium were inoculated into YPD liquid medium, followed by culture overnight at 30 °C (yeast forms). To prepare hyphae, yeast cells were cultured in the indicated medium at 37 °C in cell culture plates. To harvest the germinating hyphae at a similar length, the incubation time in each medium was different. 3 h in RPMI-based medium, 2 h 30 min in SC-based medium, and 2 h in YEP-based medium were used with $CO_2$. 2% GlcNAc replaced glucose in the original media when GlcNAc was supplied.

## *C. albicans* strains
*C. albicans* WT strain SC5314 and *ARG4* gene-transformed SN250 were used as WT strains. The mutant strains, including $hwp2^{\Delta/\Delta}$, $sap6^{\Delta/\Delta}$, $sap5^{\Delta/\Delta}$, $ece1^{\Delta/\Delta}$, $hyr1^{\Delta/\Delta}$, $rbt4^{\Delta/\Delta}$, $idh1^{\Delta/\Delta}$, $nrg1^{\Delta/\Delta}$, and $brg1^{\Delta/\Delta}$ were from the Noble collection[67] and transformed with *ARG4*. The $flo8^{\Delta/\Delta}$ mutant is described previously[68]. The mutants including $ume6^{\Delta/\Delta}$, $hgc1^{\Delta/\Delta}$, and $efg1^{\Delta/\Delta}$ are described previously[69,70]. The plasmid pENO1-BFP-NAT was a gift from Hidde Ploegh (Addgene plasmid #50046; http://n2t.net/addgene:50046; RRID:Addgene_50046)[52]. Linearized pENO1-BFP-NAT was transformed to SN250 and the $als1^{\Delta/\Delta}als3^{\Delta/\Delta}$ mutant resulting in the WT-BFP and $als1^{\Delta/\Delta}als3^{\Delta/\Delta}$-BFP.

Gene editing by CRISPR/Cas9 technology optimized for *C. albicans* was used as previously described[71] to generate the $als3^{\Delta/\Delta}$, $hwp1^{\Delta/\Delta}$, $als1^{\Delta/\Delta}als3^{\Delta/\Delta}$, $als3^{S170Y}$, $als1^{\Delta/\Delta}als3^{S170Y}$, $als3^{S170Y}als5^{\Delta/\Delta}$, $als1^{S170Y}als3^{S170Y}als5^{\Delta/\Delta}$ ($als1^{SY}als3^{SY}als5^{\Delta/\Delta}$ in short), $als3^{GPI\Delta/\Delta}$::7x His, $als3^{S170YGPI\Delta/\Delta}$::7x His, and the complementary strain $als3^{\Delta/\Delta}$::*ALS3*. Oligonucleotides are listed in Supplementary Data 1. A restriction site was introduced into the point mutation site, which was used to identify the correct mutation after PCR. In addition to digestion, the PCR product was further confirmed by sequencing.

## Quantitative RT-PCR
RNA samples were prepared using a Zymo research's Quick-RNA™ Miniprep Kit. cDNA was synthesized using a Bio-Rad iScript Reverse Transcription Kit and 500 ng total RNA. Quantitative PCR was performed using Bio-Rad SYBR Green mix on a Bio-Rad iCycler. Gene expression was normalized to the housekeeping gene *ACT1* according to the Pfaffl method[72]. Primers used are listed in Supplementary Data 1.

## Production and purification of Als3-His or Als3$^{S170Y}$-His
Als3-His and Als3$^{S170Y}$-His proteins were purified from the supernatant of $als3^{GPI\Delta/\Delta}$::7x His and $als3^{S170Y GPI\Delta/\Delta}$::7x His mutant respectively. Overnight YPD cultures were washed and diluted 1:100 into RPMI 1640 medium and cultured for 36–48 h at 37 °C, 200 rpm. The supernatants were collected, after which the Als proteins were affinity purified by passing the supernatant over a Ni-NTA column (Qiagen). The purity of each protein was verified by SDS-PAGE, showing one single band.

## Generation of BMDMs and BMDCs
BMDM derivation was carried out as described[73]. Male and female mice at the age of 6–12 weeks were used for the generation of BMDMs and BMDCs. Macrophages were derived by culturing bone marrow cells in DMEM medium (Gibco) containing 10% FBS (Corning), 1× penicillin-streptomycin (Gibco) (referred to as complete DMEM medium) with 20 ng/mL M-CSF (R&D). Medium with fresh M-CSF was added every other day. M0 macrophages were harvested on day 7. For M1 macrophages, at day 5 or 6, the medium was replaced with a complete DMEM medium with 20 ng/mL murine GM-CSF (Peprotech) and further cultured for 24–36 h[34]. BMDMs were gently washed with PBS before harvesting from plates.

BMDCs were carried out as described[74]. Bone marrow cells were placed in Petri dishes with RPMI 1640 (Gibco) supplemented with 10% FBS (Corning), 1% penicillin-streptomycin (GIBCO), 1% sodium pyruvate (Gibco), and 10 ng/mL GM-CSF (Peprotech). Cells were incubated at 37 °C with 5% $CO_2$ for 7 days. Medium was replaced every other day. On day 7, cells in suspension were collected.

## Infection or stimulation of BMDMs and BMDCs
For the measurement of cytokines, *C. albicans* were seeded at $1 \times 10^5$ cells per well in a 96-well round bottom plate and cultured at 37 °C. BMDMs or BMDCs were added to the plate with hyphae at MOI 1. If not indicated, BMDMs were infected with hyphae for 5.5 h in RPMI plus 10% FBS for ELISA or RPMI plus 0.5% FBS for immunoblotting, and BMDCs were incubated with purified Als3 (15 μg ml$^{-1}$) for 24 h. The cultures were centrifuged at $250 \times g$ for 2 min before incubation. For Western blot analysis, the infections were scaled up into 48-well plates. For inhibitor studies, the following compounds were added at the same time with infection: soluble β-glucan (InvivoGen, tlrl-wgps, 1 μg/mL), latrunculin B (Abcam, ab144291-1mg, 10 μM), cytochalasin D (Sigma-Aldrich, 504776, 2 μM), and β-methylglucoside (Sigma-Aldrich, M0779-5G, 10 mM).

## Immunoblotting
Hyphae were induced with the indicated medium in plates, and then BMDMs were added into each well of the plates at MOI 1 and cultured at 37 °C with 5% $CO_2$ for 5.5 h. After co-incubation, cells were lysed in 2x

SDS loading buffer supplemented with protease inhibitor mini cOmplete (Roche), phosphoSTOP (Roche), and 1 mM PMSF. Samples were run on SDS-PAGE gels and the gel was transferred to PVDF membranes (Millipore, IPVH00010). Non-specific binding was blocked by incubation with 5% skim milk; then membranes were incubated with the following primary antibodies: anti-IL-1β (R&D Systems, AF-401-NA, 1:2000), anti-caspase-1 (AdipoGen, AG-20B-0042--C100, 1:1000), anti-GSDMD (Abcam, ab209845, 1:1000), anti-β-actin-HRP (CST, 5125 S, 1:5000), anti-p-Syk (CST, 2710 S, 1:1000), anti-Syk (CST, 2712 S, 1:2000), anti-CD18 (CST, 47598 S, 1:1000), anti-p-p38 (CST, 9211 S, 1:1000), anti-his-HRP (CST, 9991 S, 1:2000), and anti-CD11b (Novus, NB110-89474, 1:1000). Membranes were then washed and incubated with the appropriate horseradish peroxidase (HRP)-conjugated secondary antibodies (Jackson Immuno Research Laboratories, 1:7500) for 30 min, including anti-rabbit (111-035-003), anti-mouse (115-035-003), or anti-goat (705-035-003), if HRP was not conjugated with the primary antibodies. Protein bands were visualized using Clarity or Clarity Max Western ECL Substrates (Bio-Rad), and membranes were developed with a Fujifilm LAS-4000 Imager; images were analyzed with ImageJ (1.51j8).

### Staining assays

Hyphae were induced in the indicated medium on 8-well chamber slides (Thermo Scientific). Phagocytosis assays were derived from a previously described method[44]. Macrophages were added onto hyphae in chamber slides at MOI 1, and the slides were briefly centrifuged. Phagocytosis was allowed for 2 h at 37 °C with 5% CO$_2$. Hypha-macrophage co-cultures were fixed with 4% formaldehyde for 20 min, followed by extensive washing in PBS. After phagocytosis, external *C. albicans* were labeled for 20 min at room temperature using a solution of 5 μg/mL Alexa Fluor 647-conjugated concanavalin A (Invitrogen). Then cells were permeabilized with 0.1% Triton X-100 for 5 min and stained with 10 μg/mL calcofluor white (Fluorescent Brightener 28; Sigma-Aldrich). Images were obtained on a Leica SP8 confocal microscope. β-glucan staining is described previously[73].

### Immunofluorescence

Hyphae were induced on 8-well chamber slides before the addition of BMDMs at MOI 1, brief centrifugation to ensure BMDMs contact with the hyphae, and incubation for indicated time at 37 °C with 5% CO$_2$. Cells were then washed with chilled PBS, fixed with 4% formaldehyde for 20 min, permeabilized with 0.5% Triton X-100 for 5 min, and blocked in TBS containing 10% FBS for 10 min at room temperature. Cells were stained with unconjugated primary antibodies at 4 °C overnight as follows: rabbit anti-p-Syk (CST, 2710 S, 1:100); mouse anti-pTyr (CST, 9411 S, 1:1600); rat anti-CD11b (BD Biosciences, M1/70, 1:100); rat anti-Dectin-2 (Bio-Rad, MCA2415EL, 1:100). Cells were then washed and incubated with secondary antibodies for 30 min as follows: FITC-conjugated anti-mouse (Jackson ImmunoResearch); AlexaFluor488-conjugated anti-rabbit (Invitrogen); AlexaFluor568-conjugated anti-rat (Invitrogen); AlexaFluor647-conjugated anti-rat IgG2a (Biolegend), and eFluor 570-conjugated anti-rat IgG2b (Invitrogen). Phalloidin-iFluor 633 Reagent (Abcam) was used to stain the F-actin. Slides were examined with a Leica TCS SP8 confocal microscope. Image analysis was performed using Leica LAS AF software.

### Biochemical studies to detect the interaction of Als3 with CD11b

For the detection of pull-down CD11b with hyphae, mouse iBMDM cells (a generous gift from Kate A. Fitzgerald) were incubated with hyphae in RPMI medium for 45 min at 37 °C with 5% CO$_2$. Cells were then washed with cold PBS including 1 mM Mg$^{2+}$ and 1 μM Ca$^{2+}$. To stabilize the interaction, DSP (ProteoChem) was added at a final concentration of 1.5 mM and reacted at room temperature for 30 min. The reaction was quenched for 15 min by addition of 20 mM Tris, PH7.5. The cells were lysed in cold lysis buffer, TBS with 1% N-octyl-β-D-glucopyranoside (β-OG, Sigma), EDTA-free protease inhibitors (Pierce), and 1% BSA for 30 min at 4 °C with rotation. Then, the hyphae were washed with lysis buffer 6 times. The boiled samples in 2x SDS loading buffer were subjected to SDS-PAGE and blotted with CD18 antibody.

Pull-downs with iBMDM lysates were also performed with His-tagged Als3 or His-tagged Als3$^{S170Y}$. The iBMDM cells were resuspended in cold lysis buffer as above containing 2 mM MnCl$_2$ for 30 min at 4 °C with rotation. The samples were centrifuged at 1900 × g, 4 °C for 5 min, and the supernatant containing the solubilized portion was collected. The solubilized portion (from 1.25 × 10$^6$ cells) was incubated with 20 μg of Als3-His or Als3$^{S170Y}$-His incubated with TBS for 30 min at 4 °C with rotation. CD11b mAb M1/70 (BD sciences, 1:50) or polyclonal antibody (Novus, 1:100) was added to the lysates. The samples were incubated with 45 μl of Dynabeads protein A (Invitrogen) for 3 h at 4 °C with rotation. The resin was washed with lysis buffer with MnCl$_2$, and the proteins were eluted with TBS containing 1% β-OG and 10 mM EDTA. The samples were boiled with SDS loading buffer and fractionated by 6% SDS-PAGE. The gels were transferred to PVDF membrane. The membranes were blotted with a-His-HRP antibody (CST, 9991 S, 1:2000).

For the pull-down with purified Als3-His or Als3$^{S170Y}$-His and purified mouse CR3, 0.4 μg recombinant mouse CR3 (Integrin α$_M$β$_2$, R&D Systems) was incubated with 2 μg of purified Als3-His, Als3$^{S170Y}$-His, or PBS in the presence of 0.1% β-OG and 1% BSA with or without Mn$^{2+}$ and fibronectin (Thermo Scientific, AAJ64738MCR) for 30 min at 4 °C with rotation. The samples were incubated with polyclonal antibody CD11b (Novus) at 1:100 and 25 μl Dynabeads protein A, washed and eluted in TBS buffer containing 1% β-OG buffer, 10 mM EDTA, and 200 mM DTT. The immunoprecipitated proteins were collected by boiling in 2x SDS loading buffer at 100 °C for 5 min.

For particulate β-glucan pull-down assays, 0.25 μg recombinant human CR3 (Integrin α$_M$β$_2$, R&D Systems) and 5 μg of purified Als3-His was incubated with 250 μg particulate β-glucan (InvivoGen, tlrl-wgp) in PBS buffer with 2 mM Mn$^{2+}$ for 2 h at room temperature with rotation. 1 mg ml$^{-1}$ soluble β-glucan, 1 mg ml$^{-1}$ β-methylglucoside, 2 μg monoclonal antibody ICRF44 (BioLegend, 301302), or its isotype control (BioLegend, 400101) was used for competition. The samples were washed and eluted in PBS buffer containing 10 mM EDTA. The eluates were separated with 6.5%-15% SDS-PAGE.

### Immunoprecipitation of activated CD11b

BMDMs were incubated with hyphae in RPMI medium for 45 min at 37 °C with 5% CO$_2$. Cells were then washed with cold PBS including 1 mM Mg$^{2+}$ and 1 μM Ca$^{2+}$. To stabilize the interaction, DSP was added at a final concentration of 1.5 mM. The reaction mixture was incubated at room temperature for 30 minutes. 20 mM Tris, PH7.5, was added and sat for 15 min to stop the reaction. The cells were lysed in cold lysis buffer, TBS with 1% β-OG, and EDTA-free protease inhibitors for 30 min at 4 °C with rotation. Anti-CD11b (Invitrogen, 14-0113-81, clone CBRM1/5, 1:25) and Dynabeads protein A were used to pull down activated CD11b. The washed beads were boiled with 2x SDS loading buffer, and then the supernatants were subjected to SDS-PAGE and blotted with a polyclonal CD11b antibody (Novus, NB110-89474).

### In vivo systemic *Candida* infections

*C. albicans* strains were cultured in YPD broth, and grown at 30 °C, 200 rpm. Yeast cells were washed in PBS, renumerated, and injected intravenously via the lateral tail vein. Animals were randomly assigned to the different groups. Researchers were not blinded to the experimental groups because the endpoints (fungal burden and cytokine levels) were objective measures of disease severity. To determine organ fungal burden, mice were sacrificed, after which the kidneys and brain were harvested, weighed, homogenized, and quantified on YPD.

## ELISA

Co-culture supernatant was collected, and cytokines were measured by mouse IL-1β ELISA Ready-Set-Go cytokine kits (eBioscience).

## SYTOX green assay

BMDMs were resuspended at $1 \times 10^6$ cells/mL with 1 μM Sytox Green (Invitrogen) and plated onto hyphae at MOI 1 in 96-well plates at $1 \times 10^5$ BMDMs per well. Fluorescence (485 Ex/525 Em) was quantified every 15 min for 16 h using a Varioskan LUX Plate Reader (Thermo Scientific) at 37 °C with 5% $CO_2$.

## Lactate dehydrogenase (LDH) assay for cell death

LDH release was measured to analyze cell death using a CytoTox 96® Non-Radioactive Cytotoxicity Assay kit (Promega) according to the manufacturer's instructions.

## Statistics and software

Statistical analysis was determined using indicated methods with GraphPad Prism software. Error bars indicate mean ± SEM and $P$ values less than 0.05 were considered significant. BioRender was used to illustrate figures. Figure 7 was adapted from "Outside-in and Inside-out Integrin Signaling Pathways", by BioRender.com (2024), retrieved from https://app.biorender.com/biorender-templates.

## Reporting summary

Further information on research design is available in the Nature Portfolio Reporting Summary linked to this article.

## Data availability

The source data for Fig. 1a–f; 2a, b; 3a–c; 4a–e; 5a–f; and 6a–d, g and Supplementary Figs. 1a–d; 2a, b; 3a, b; 4a, b; 5a, b; and 6a–f are provided as a Source Data file in this paper. Source data are provided with this paper.

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

## Acknowledgements

We thank Dr. Katherine A. Fitzgerald for iBMDM. This research is supported by NIH GM117111 (HL), NIH R01EY18612 (EP), and NIH R01DE022600 (SGF).

## Author contributions

T.T.Z. and H.L. conceived and designed the study; T.T.Z. performed most of the experiments and data analysis; N.V.S. and S.G.F. performed in vivo experiment; M.M. and E.P. assisted with getting bone marrows and other reagents; Q.Y. purified fungal proteins; R.G. assisted with CRISPR-Cas9 experiment; M.Y. assisted with cell cultures; T.T.Z. wrote the manuscript; T.T.Z., H.L., S.G.F. and E.P. reviewed and edited the manuscript; H.L., E.P. and S.G.F. secured the funding.

## Competing interests

The authors declare no competing interests.
