## [Peer Review File · Nature Communications]

Hyphal Als proteins act as CR3 ligands to promote immune responses against *Candida albicans*REVIEWER COMMENTS

Reviewer #1 (Remarks to the Author):

CR3-deficient mice rapidly succumb to infection by *Candida albicans*. This manuscript describes Bone Marrow Derived Macrophage response to *C. albicans* hyphae. The interaction is consistent with response being mediated through integrins CR3, (alias CD11b/CD18), and/or CR4 (CD11c/CD18). The work shows that macrophage responses to hyphae include IL-1 β secretion and inflammasome induction leading to macrophage death. In normal mice, this macrophage response triggers inflammatory reactions and acquired immunity to clear the infection.

I summarize the major findings:

1. Deletions of *C. albicans* ALS3 induced attenuated responses, and also increased kidney burden 4-6 logs in infected mice (Fig. 1). Als1 and Als5 may also be minor contributors to the response.
2. Als3 expression was induced 2-3 fold by addition of GlcNAc to fungal growth media. The level of Als3 expression correlated to the amount of the active forms of Caspase-1 and IL-1 β (Fig. 2). An als5 Δ/Δ fungal strain that expresses Als1S170Y and Als3S170Y (hereafter "alstriple") was the most attenuated, ~70% in Caspase-1 response.
3. Fig. 3 demonstrates that generation of active forms of IL-1 β , Casp-1, and the pyroptosis mediator gasdermin are attenuated ~50%-60% in BMDMs from CD18 $^{-/-}$ mice. Response was further attenuated if alstriple fungi were the inducers, or by treatment with an antibody to CD11b. Recombinant Als3 induced secretion of IL-1 β , but recombinant Als3S170Y was ineffective. Phosphorylation of Syk was prolonged in response to alstriple, and the prolongation was dependent on CD18 and dectin-1.
4. Hyphae bound CD18, and anti-CD11b pulled down Als3, but not Als3S170Y (Fig. 4a,b). BMDM Casp-1 and IL-1 β responses were also attenuated in the presence of EDTA or soluble glucans. This result implies that binding of yeast glucan is also necessary for maximal response.
5. Fig. 5 shows that phosphorylation of Syk in unprimed M0 BMDMs was dependent on presence of WT Als3 on the hyphae, as well as both CD18 and dectin-2 in the macrophages. Anti-CD11b inhibited Syk phosphorylation. In WT BMDMs, p-Tyr, CD11b, and F-actin colocalized to a collar surrounding a hypha, But not in CD18 $^{-/-}$ BMDMs or with hypha from als1 Δ/Δ als3 Δ/Δ *C. albicans*. M1 BMDMs showed colocalized p-Syk, CD11b, and Dectin-2.
6. Fig. S1. Mphage death was delayed in als3 Δ/Δ or efg1 Δ/Δ (EFG1 is necessary for hyphal formation in *C. albicans*). Adding deletions als1 Δ/Δ als5 Δ/Δ does not affect the result significantly. alstriple shows little activation of Casp-1
7. Fig. S2 ALS1 expressed well in YEP GlcNAc or RPMI; Als5 expression is indifferent to medium.
8. Fig. S3a. Deletion of CD18 $^{-/-}$ DOES NOT affect LDH release; alstriple 70% of WT LDH secretion. b: CD18 $^{-/-}$ gives 50% less IL-1 β . Als3SY is inactive in IL-1 β release. alstriple shows more extracellular hyphae, implying less endocytosis.
9. Fig. S4 recombinant proteins: anti CD11b pulls down WT Als3, not Als3SY.
10. Fig. S5. Syk phosphorylation increased by Mg $^{++}$, abrogated by EDTA; CD11b activation abrogated in CD18 $^{-/-}$, dectin-2 $^{-/-}$

Overall, the paper demonstrates that BMDM's respond to hyphae by secreting IL-1 β and activating Caspase-1 to induce pyroptosis. These responses depend in part on CD18 and CD11b integrin subunits. Also important is glucan recognition by Dectin-2. Response also includes localization and phosphorylation of Syk. These are definitive steps in macrophage early response to *C. albicans* and may indeed be a critical determination of immune response to an extremely important pathogen. This level of detail in elucidation of such a mechanism is unprecedented in mycology.

However, there are several issues that need to be addressed, either experimentally or by revision of the manuscript. In approximate order of importance they are:

1. There is only 50% attenuation of responses in most of the reported data? Is there another pathway? The possible participation of other Als proteins should be acknowledged as a possibility. More troubling is the residual activity in CD18 $^{-/-}$ and dectin2 $^{-/-}$. Again, is there involvement of paralogous receptors?
2. Evidence for involvement of CD18 lectin domain is extremely weak. No direct evidence for involvement of this domain is reported in the manuscript. Rather, the evidence of inhibition by glycans also applies to dectins. I could find no report that fibronectin is an inhibitor of glycan

binding to the integrin lectin domain. (Fibrinogen is an inhibitor.). Also, the binding of Als3 to activated CD11b is abrogated in a dectin-2 deletion, supporting the critical role for dectin-2, rather than the lectin domain of CR3.

3. Involvement of p-Syk is inferred from literature. The evidence is convincing that Syk is localized to the right place and is phosphorylated in response to Als3-CR3/4 binding. However, this manuscript does not show direct evidence that Syk phosphorylation is necessary for IL-1 β or Caspase-I activation.

4. There is no effect of the als deletions on BMDM release of LDHm, and LDH release is high also in CD18 $^{-/-}$. These are puzzling results given the high percentage of cell death. What is the explanation?

5. Minor concerns: A number of terms and reagents are not fully described:

a. iBMDM

b. Origins or design of the Als mutant proteins and their secreted forms: "to be described elsewhere" or "manuscript in preparation" is not sufficient.

c. What are the dotted lines in fig 1b and 1c?

d. Lines 123-4 "The amount of fungal-induced cell death...independent" Insert "was"

e. Lines 130-135. What is the rationale for use of dendritic cells in this experiment? How are they related to BMDMs?

f. Lines 398-9: Correct syntax in sentence "Incubate the reaction mixture at room temperature for 30 min. 20 mM Tris, PH7.5 was added and sat for 15 min to stop the reaction."

g. Fig. 1e. Is the "als3" strain als3 Δ/Δ ?

h. Fig. 4: Cytochalasin D appears to have no effect. Is that true?

i. Line 571 "Elutes" should be "Eluates."

Reviewer #2 (Remarks to the Author):

In the manuscript, the authors show that *Candida albicans* hyphae Als3 binds to the I-domain of CR3, which is essential for NLRP3 activation and IL-1 β production. Macrophages with CR3 deficiency fail to initiate Als3-induced immune response. This manuscript tells a very interesting story about the interaction between the host and fungal pathogen. However, due to inadequate experimental evidence and confused description, the conclusion that "Als proteins act as CR3 ligands to promote immune responses against *C. albicans* remains farfetched. Some points are addressed below.

Major points:

Line 57:

Despite the fact that many inflammatory cytokines (IL-1 β , IL-6, TNF- α , IL-12 etc.) are important for host immune defense against *C. albicans* infection, the authors focus on the NLRP3 inflammasome activation and IL-1 β production. Does Als proteins facilitate other cytokines production?

Line 83:

The unpublished data cannot serve as reference. Besides, it is confusing that the authors use als3 single deletion strain or triple mutant strain irregularly. If the authors try to demonstrate the biological function of Als protein family, experiments about other single deletion or mutant strain are needed. If the authors focus on the Als3 induced immune response, compare the difference between Als protein family and select the most reasonable object to clarify in the context.

Figure 1a

To conclude "Live elongating hyphae induce higher level of IL-1 β compared to yeast", the hyphae-locked strain such as nrg1 or tup1 are needed for comparison. Besides, the schematic diagram is not needed since it cannot provide valuable information.

Figure 1c and Supplementary Figure 1a

References or experimental evidences are required to illustrate the relationship between macrophage death and IL-1 β production.

Fig. 1d and Supplementary Figure 1c

It is not a consensus that the complemented strain can regard as the control, the WT strain SN250 is still required.

Figure 1e

The CFU in kidney and brain is not enough to support the conclusion that "ALS3 is required for IL-

1beta production and fungal clearance". Does recombinant mIL-1beta can rescue the phenotype induced by als3 deletion strain? The authors must detect the IL-1beta production, T cell response and the activation of innate immune cells between als3 deletion and ALS3 complemented strain infected mice.

Figure 2a

The authors' data show that the Als3 level is closely correlative with beta glucan. Does als3 influence the exposure of beta glucan in the surface of *C. albicans*? Whether als3 deletion strain induced decreased macrophage response shown in figure 1 is due to decreased beta glucan in fungi? This question also exists in Figure 3, 4 and 5. Purified hyphae Als3 are needed in all in vitro experiments.

Figure 2b&c

The experimental design lacks rigorous control and the annotation makes audience confused. As the authors pronounced that the most important factor that inducing als3 expression in the fungal culture medium shown in figure 2b is GlcNAc, why did the authors compare the RPMI and SC in figure 3c? Besides, the annotation in the top of figure 2b would likely to tell audience the RPMI is GlcNAc free, which make it difficult to understand what the authors wish to convey.

Figure 3a

The authors use CD18 deficient BMDM to descript Als3 induced inflammasome activation is CR3 dependent. However, triple mutant strain induced IL-1beta and caspase-1 activation still significantly lower than complemented strain in CD18 deficient BMDM, which suggests that Als3 induced inflammasome activation may be CR3 independent.

Figure 3c

The switch between BMDM and BMDC is confusing since it is not well established that BMDC cannot engulf fungus.

Figure 3d

It is not convincing to say that CR3 promotes als-mediated inflammasome activation, since the termination of p-Syk cannot serve as the evidence of activation of CR3 signaling pathway. More work about the Als3-CR3 axis must be provided. Besides, we do not encourage researchers to compare bands in different membranes.

Figure 4

The biochemical method the author used is not solid enough to prove the author's statement in abstract that "CR3 I domain recognize Als family proteins". More in vitro binding assay include ELISA and immunofluorescence using purified Als and CR3 mutant should be performed to clarify the synergize effect between Als3 and beta-glucan. Previous studies can provide effective guidance (PMID: 37872182).

Figure 5a

The *C. albicans* induced syk phosphorylation in CD18 deficient BMDMs are totally different with authors shown in figure 3d. This is paradoxical even if macrophage polarization may influence.

Figure 5d

It is hardly to observe the decreased clustering of p-Tyr and F-actin, nor the difference of F-actin cuff formation between CD18 deficient and WT BMDMs. The fluorescence background seems not normalized neither. The data shown in figure 5d barely match the statement "Als promoting F-actin remodeling" in abstract. More significant images and quantification of protein co-localization are needed.

Figure 5e&f

The direct connection between Als3 induced p-syk and inflammasome activation are not well illustrated. The Src family kinase downstream of beta-integrin play an important role in F-actin polymerization and signaling transduction, authors should test whether Src is involved in Als3 induced inflammasome signaling. In addition, as the authors mentioned F-actin remodeling, it is most likely to influence the fungal phagocytosis, which has a great effort in "inside-out signaling" and immune response. The authors should provide that whether the macrophage phagocytosis is different to als3 delete and control strain.

Reviewer #3 (Remarks to the Author):

This paper examines a previously uncharacterized interaction between ALS3 and host molecules involved in the immune response and I think the work will be of interest to the field.

Comments

General

There are several references to precursor information that clearly led to this work that will be included in another manuscript in preparation for publication. It's unfortunate that submission of this paper didn't wait for those results to be published.

Overall the authors assume that the reader is familiar with the different sizes of inactive and active forms of the proteins examined by western blot. A little more information would make the results clearer to the non-immunologist.

There are several places where the descriptions are more casual than might be expected in a paper (like a notable surge). It would be better to use more formal and more accurate descriptions.

The rationale for why particular mutant strains were used in particular experiments isn't always clear and there is inconsistency between using specific strain names vs. wild-type throughout the text and the figure labels.

Results

Figure 1. The *flo8* mutant mentioned in figure 1a is not in the strain list. I think the 6 well plate cartoon could be omitted as it doesn't illustrate anything meaningful. Panels b and c have WT and SN250, respectively. Was the same strain used in both? What is the significance of the dashed line in panels b and c and why is it in a different location in each? The legend mentions unstimulated macrophages but the graphs are labelled uninfected.

Lines 84/85. If the result wasn't significantly different, just say that instead of emphasizing differences.

Line 94. It may be helpful to readers to describe these as low dose infections since they are substantially below what is used in a typical tail vein injection when testing virulence.

Figure 2. Why is a different wild-type strain being used here than in Figure 1 or Figure 3?

Line 102-105. Is this trying to say that GlcNAc was chosen because of a known effect on beta glucan exposure and that was a condition of interest? As written it seems to imply that GlcNAc in particular induces hyphae while other media do not, which isn't accurate, and it doesn't make clear the choice of GlcNAc over other media that also induce the expression of hypha specific genes.

Line 112. The result is significant or not. It can't be more significant than something else.

Lines 115-117. This description is confusing and seems more like discussion than results. Stating that Als proteins from hyphae are doing something makes it sound like Als proteins were isolated from hyphae and then used in the experiment.

Line 121. The text refers to wild-type, but Fig 3a is labelled Control, which elsewhere refers to a complement strain. What strain was used?

Figure 4. Why does panel b use Mn but panel c uses Mg?

Lines 176-178. "Moreover, EDTA treatment diminished the priming signaling induced by hyphae, as evidenced by the low pro-IL-1 β level, and the inflammasome signaling induced by Als proteins (Fig. 4c)." The part of this is clear in the figure. What is the second part of the statement referring to?

Line 186/187. This sort of statement seems a bit out of place in a results section and certainly shouldn't be a single sentence paragraph.

Lines 199/204. The levels of p-Syk are highest in SN250, but the pattern is the same in all three strains (upper panel). The last sentence of the paragraph seems redundant.

Line 221. Are the methodological details like reference to beads necessary?

Line 230. It might be clearer to describe the results, not the conclusion.

Discussion

Lines 263-272.

The importance of the comparison being made here isn't clear. Induction in one medium in vitro looked much like another, and all were similar to in vivo induction. The cited work examined cells grown in shaking broth cultures so the comparison to the work presented here with cells grown in dishes isn't direct.

The comments about the link to cancer feel tacked on and even as a speculation reaching too far.

Language

Many sentences that begin with an adverb, like notably in line 201, which could be omitted. Throughout, growing cells in broth in a well of a multi-well dish cannot be described as growth on a plate, which would instead refer to on a solid medium.

Line 51 might be better as "Pra1, whose production is higher in hyphae"

Line 122. Reduced or lower would be better than diminished. Restructuring the sentence might make the whole statement clearer.

Line 123/124. This statement needs a verb.

Line 153. Fix inline citation.

Line 159/160. "Our results demonstrated that" is unnecessary.

Line 189. Cut the first sentence of the paragraph.

Line 163/164. Rephrase to remove the redundancy of analyzing interaction by investigating interactions.

Line 212-217. Rephrase to remove redundancy.

Line 225 should read phosphatases.

Line 241/242 should read "recognize" and "promote".

Line 257-260. The statement starting Furthermore is a run-on sentence and needs to be reworked.

Line 284-286. Please rephrase these awkward sentences.

REPLY TO REVIEWER COMMENTS

We thank the reviewers for their constructive comments and have addressed them as described below. The major changes to the revised manuscript are:

Major new data are pull-down assays with particulate β -glucan, purified Als3, and recombinant CR3 that show the synergy between CD11b I domain and lectin-like domain and Als3 binding to the CD11b I domain (using I domain-specific antibody ICRF44) (Fig. 5f and 5d).

Additional points addressed with new experimental data:

- (1) involvement of wild-type (SN250) and other *als* mutants (supplemental figure 1d, and Fig. 2b);
- (2) Additional cytokines affected by the *als* mutant in infected kidneys (new Fig. 2b and new supplementary Fig. 2b);
- (3) Additional hyphal mutants showed reduced IL-1 β release (supplemental figure 1a).
- (4) Syk phosphorylation in C57BL/6 and *Dectin-2*^{-/-} M0 BMDMs infected with purified Als3 or Als3^{S170Y} (Supplemental figure 6b).

All author responses are in red.

Reviewer #1:

CR3-deficient mice rapidly succumb to infection by *Candida albicans*. This manuscript describes Bone Marrow Derived Macrophage response to *C. albicans* hyphae. The interaction is consistent with response being mediated through integrins CR3, (alias CD11b/CD18), and/or CR4 (CD11c/CD18). The work shows that macrophage responses to hyphae include IL-1 β secretion and inflammasome induction leading to macrophage death. In normal mice, this macrophage response triggers inflammatory reactions and acquired immunity to clear the infection.

I summarize the major findings:

1. Deletions of *C. albicans* ALS3 induced attenuated responses, and also increased kidney burden 4-6 logs in infected mice (Fig. 1). Als1 and Als5 may also be minor contributors to the response.
2. Als3 expression was induced 2-3 fold by addition of GlcNAc to fungal growth media. The level of Als3 expression correlated to the amount of the active forms of Caspase-1 and IL-1 β (Fig. 2). An *als5* Δ/Δ fungal strain that expresses Als1S170Y and Als3S170Y (hereafter “als triple”) was the most attenuated, ~70% in Caspase-1 response.
3. Fig. 3 demonstrates that generation of active forms of IL-1 β , Casp-1, and the pyroptosis mediator gasdermin are attenuated ~50%-60% in BMDMs from CD18^{-/-} mice. Response was further attenuated if alstriple fungi were the inducers, or by treatment with an antibody to CD11b. Recombinant Als3 induced secretion of IL-1 β , but recombinant Als3S170Y was ineffective. Phosphorylation of Syk was prolonged in response to alstriple, and the prolongation was dependent on CD18 and dectin-1.
4. Hyphae bound CD18, and anti-CD11b pulled down Als3, but not Als3S170Y (Fig. 4a,b). BMDM Casp-1 and IL-1 β responses were also attenuated in the presence of EDTA or soluble

glucans. This result implies that binding of yeast glucan is also necessary for maximal response.

5. Fig. 5 shows that phosphorylation of Syk in unprimed M0 BMDMs was dependent on presence of WT Als3 on the hyphae, as well as both CD18 and dectin-2 in the macrophages. Anti-CD11b inhibited Syk phosphorylation. In WT BMDMs, p-Tyr, CD11b, and F-actin colocalized to a collar surrounding a hypha, But not in CD18^{-/-} BMDMs or with hypha from *als1Δ/Δ als3Δ/Δ C. albicans*. M1 BMDMs showed colocalized p-Syk, CD11b, and Dectin-2.
6. Fig. S1. Mphage death was delayed in *als3Δ/Δ* or *efg1Δ/Δ* (EFG1 is necessary for hyphal formation in *C. albicans*). Adding deletions *als1Δ/Δ als5Δ/Δ* does not affect the result significantly. *alstrip* shows little activation of Casp-1
7. Fig. S2 ALS1 expressed well in YEP GlcNAc or RPMI; Als5 expression is indifferent to medium.
8. Fig. S3a. Deletion of CD18^{-/-} DOES NOT affect LDH release; *alstrip* 70% of WT LDH secretion. b: CD18^{-/-} gives 50% less IL-1β. Als3SY is inactive in IL-1β release. *alstrip* shows more extracellular hyphae, implying less endocytosis.
9. Fig. S4 recombinant proteins: anti CD11b pulls down WT Als3, not Als3SY.
10. Fig. S5. Syk phosphorylation increased by Mg⁺⁺, abrogated by EDTA; CD11b activation abrogated in CD18^{-/-}, dectin-2^{-/-}

Overall, the paper demonstrates that BMDM's respond to hyphae by secreting IL-1β and activating Caspase-1 to induce pyroptosis. These responses depend in part on CD18 and CD11b integrin subunits. Also important is glucan recognition by Dectin-2. Response also includes localization and phosphorylation of Syk. These are definitive steps in macrophage early response to *C. albicans* and may indeed be a critical determination of immune response to an extremely important pathogen. This level of detail in elucidation of such a mechanism is unprecedented in mycology.

However, there are several issues that need to be addressed, either experimentally or by revision of the manuscript. In approximate order of importance they are:

1. There is only 50% attenuation of responses in most of the reported data?

Is there another pathway?

The modified description of the redundancy of inflammasome is in the Introduction (line 62-70) “The NLRP3 inflammasome can be activated by diverse stimuli and cellular events. β-glucans, the most abundant cell wall component of fungi, are recognized by cell surface receptors, such as Dectin-1 and CR3, and cytoplasmic inflammasome components, resulting in proinflammatory gene transcription and NLRP3 inflammasome activation^{26, 27, 28}. Hyphae activate the NLRP3 inflammasome significantly more than yeasts, leading to increased production of IL-1β^{29, 30, 31, 32}. Candidalysin and Saps also trigger NLRP3 inflammasome activation^{33, 34}.”

The possible participation of other Als proteins should be acknowledged as a possibility.

Involvement of other Als proteins are shown in the new supplemental figure 1d, Fig. 2b and supplemental figure 2b. “The *als1^{S170Y}als3^{S170Y}als5^{Δ/Δ}* (*als1^{SY}als3^{SY}als5^{Δ/Δ}* in short) triple mutant induced lower level of IL-1β compared to the *als3^{S170Y}als5^{Δ/Δ}* double mutant (Supplementary Fig. 1d).” “In comparison with mice infected with WT *C. albicans*, those infected with the *als3^{Δ/Δ}* mutant showed a trend towards reduced cytokines and chemokines, including IL-1β, TNF-α, IL-6, IL-10, CXCL1/KC, and CXCL2/MIP-2α, with only the reduction of TNF-α being statistically significant (Fig. 2b and Supplementary Fig. 2b). In contrast, the *als1^{SY}als3^{SY}als5^{Δ/Δ}*

strain induced significantly lower levels of all these cytokines and chemokines. Thus, the Als family is required for optimal immune responses.

More troubling is the residual activity in *CD18*^{-/-} and *dectin2*^{-/-}. Again, is there involvement of paralogous receptors?

In addition to *CD18* and *Dectin-2*, *Dectin-3* is also involved in hyphal recognition, which has been mentioned in the discussion. Although *Dectin-2* and *Dectin-3* are reported to function together (Zhu 2013), there remains the possibility that *Dectin-3* can function independently.

2. Evidence for involvement of *CD18* lectin domain is extremely weak. No direct evidence for involvement of this domain is reported in the manuscript. Rather, the evidence of inhibition by glycans also applies to dectins.

We have performed new pull-down assays with particulate β -glucan, purified Als3, and recombinant CR3, included in the revised Fig. 5f and 5d. Our new data support Als3 binding to the *CD11b* I domain, and there is synergy between *CD11b* I domain and lectin-like domain.

“Based on this, we performed a pull-down with particulate β -glucan as the solid phase and used soluble β -glucan or β -methylglucoside as antagonists. Als3 greatly increased CR3 binding to particulate β -glucan (Fig. 5d). Soluble β -glucan and β -methylglucoside both exhibited a partial inhibition of the Als3-mediated interaction between CR3 and particulate β -glucan (Fig. 5d). In BMDMs, we also found that hyphal Als-mediated inflammasome activation was largely inhibited by β -methylglucoside and soluble β -glucan (Fig. 5e), indicating that hyphal β -glucan is involved in the Als-mediated inflammasome signaling.”

“In the particulate β -glucan pull-down assay, the Als-mediated interaction of CR3 and β -glucan was also significantly inhibited by the I-domain specific antibody ICRF44, but not its isotype control (Fig. 5f), further supporting an association of Als3 with the *CD11b* I domain and the synergy between the I domain and the lectin-like domain in ligand binding.”

I could find no report that fibronectin is an inhibitor of glycan binding to the integrin lectin domain. (Fibrinogen is an inhibitor.).

Fibronectin is a known *CD11b* I domain ligand of CR3, and its reference (PMID: 12565824) has been included in the text.

Also, the binding of Als3 to activated *CD11b* is abrogated in a *dectin-2* deletion, supporting the critical role for *dectin-2*, rather than the lectin domain of CR3.

Yes, *Dectin-2* plays a critical role as shown in our text: “To explore the contribution of *Dectin-2* to CR3 activation, immunoprecipitation was performed with the monoclonal antibody (mAb) CBRM1/5 that recognizes an activation-specific epitope of *CD11b*^{56, 57}. Activated *CD11b* was pulled down from WT macrophages, but not from *CD18*^{-/-} or *Dectin-2*^{-/-} BMDMs after *C. albicans* hyphal infection (Supplementary Fig. 6c). Therefore, we propose that *Dectin-2* provides “inside-out signaling” for CR3 activation, thereby enhancing the binding of Als3 to the I domain of CR3.” “In *Dectin-2*^{-/-} BMDMs, Als3- or Als3^{S170Y}-induced Syk phosphorylation was undetectable (Supplementary Fig. 6b), suggesting that α -mannans of Als3 or Als3^{S170Y} contribute to *Dectin-2*-mediated signaling.”

3. Involvement of p-Syk is inferred from literature. The evidence is convincing that Syk is localized to the right place and is phosphorylated in response to Als3-CR3/4 binding. However,

this manuscript does not show direct evidence that Syk phosphorylation is necessary for IL-1 β or Caspase-1 activation.

Syk signaling has been shown to control both pro-IL-1 β gene transcription (signal 1/priming) and Caspase-1 activation (signal 2) after stimulation of unprimed M0 with *C. albicans* yeast cells (see Fig2 of PMID: 19339971). We also see a reduced level of syk phosphorylation in both CD18^{-/-} M0 macrophages or WT M0 macrophages infected with the *als* mutants (Fig. 6a, c). By contrast, in GM-CSF-primed M1 macrophages, while the control strain and the *als* triple mutant induced similar levels of p-Syk (fig. 4d) and inflammasome-independent cytokine TNF- α (Fig. 4e), the *als* mutant induced a lower level of inflammasome-dependent cytokine IL-1 β than the control *C. albicans* strain. Therefore, Als proteins have an additional function in inflammasome activation (this is revealed in GM-CSF primed macrophages), in addition to its function in Syk activation (observed in M0 macrophages).

4. There is no effect of the *als* deletions on BMDM release of LDHm, and LDH release is high also in CD18^{-/-}. These are puzzling results given the high percentage of cell death. What is the explanation?

In Fig. 7d of PMID:25063877, CR3 is required for BMDC cell death triggered by heat-killed *C. albicans*, but not by live *C. albicans*. Live *C. albicans* can induce several types of cell death, including pyroptosis, apoptosis, ferroptosis, and necroptosis. *C. albicans* hyphae has been shown to induce less GSDMD activation (an indicator of pyroptosis) in CD18^{-/-} BMDMs (Fig. 4), but it may increase other types of cell death in CD18^{-/-} BMDM.

5. Minor concerns: A number of terms and reagents are not fully described:

a. iBMDM

Modified. Thanks.

b. Origins or design of the Als mutant proteins and their secreted forms: “to be described elsewhere” or “manuscript in preparation” is not sufficient.

The description has been added. Thanks.

c. What are the dotted lines in Fig 1b and 1c?

Dashed lines are to separate results with mutant strains from control strains. This description has been added to the figure legend. Thanks.

d. Lines 123-4 “The amount of fungal-induced cell death...independent” Insert “was”

Added, thanks.

e. Lines 130-135. What is the rationale for use of dendritic cells in this experiment? How are they related to BMDMs?

In this manuscript, we used BMDCs to measure IL-1 β release in response to purified Als3 proteins at the protein concentration of 15 $\mu\text{g ml}^{-1}$, because BMDMs released a very low level of IL-1 β at the same concentration of purified Als3 protein. Based on the publication (PMID: 25063877), BMDCs tend to produce higher cytokine in shorter infection periods compared to BMDMs. In addition, BMDCs have been used to analyze *C. albicans*-induced inflammasome activation (PMID: 20401456). For these reasons, BMDCs were used in these experiments.

f. Lines 398-9: Correct syntax in sentence “Incubate the reaction mixture at room temperature for 30 min. 20 mM Tris, PH7.5 was added and sat for 15 min to stop the reaction.”

Changed into “To stabilize the interaction, DSP (ProteoChem) was added at a final concentration of 1.5 mM and reacted at room temperature for 30 min. The reaction was quenched for 15 min by addition of 20 mM Tris, PH7.5.”

g. Fig. 1e. Is the “als3” strain als3 Δ/Δ ?

Corrected, thanks.

h. Fig. 4: Cytochalasin D appears to have no effect.

We modified the text: “Our results showed that CytoD treatment weakened the difference between the control strain and the *als1^{SY}als3^{SY}als5 ^{Δ/Δ}* mutant but did not reduce the level in activating inflammasome. Lat B treatment reduced the levels of inflammasome signaling in both the control strain and the *als1^{SY}als3^{SY}als5 ^{Δ/Δ}* (Fig. 6g). ”

i. Line 571 “Elutes” should be “Eluates.”

Corrected, thanks.

Reviewer #2:

In the manuscript, the authors show that *Candida albicans* hyphae Als3 binds to the I-domain of CR3, which is essential for NLRP3 activation and IL-1beta production. Macrophages with CR3 deficiency fail to initiate Als3-induced immune response. This manuscript tells a very interesting story about the interaction between the host and fungal pathogen. However, due to inadequate experimental evidence and confused description, the conclusion that “Als proteins act as CR3 ligands to promote immune responses against *C. albicans* remains farfetched –

We have performed additional experiments to support that Als protein act as CR3 ligands to promote immune against *C. albicans*, including (1) New pull-down assays with particulate β -glucan, purified Als3, and recombinant CR3 showing Als3 binding to the CD11b I domain (using I-domain specific antibody ICRF44) and the synergy between CD11b I domain and lectin-like domain are included in the revised Fig. 5f and 5d. (2). New Fig. 2b and new supplementary Fig. 2b show that mice infected with the *als* triple mutant was defective in the production of IL-1 β , IL-6, TNF-alpha, IL-10, CXCL1/KC, and CXCL1/MIP-2 α in infected kidneys.

Some points are addressed below.

Major points:

Line 57:

Despite the fact that many inflammatory cytokines (IL-1beta, IL-6, TNF-alpha, IL-12 etc.) are important for host immune defense against *C. albicans* infection, the authors focus on the NLRP3 inflammasome activation and IL-1beta production. Does Als proteins facilitate other cytokines production?

As shown in the new Fig. 2b and new supplementary Fig. 2b, we measured many cytokines and chemokines and found that mice infected with the *als* mutant had decreased production of IL-1 β , IL-6, TNF-alpha, IL-10, CXCL1/KC, and CXCL1/MIP-2 α in their kidneys. The *als* triple mutant

is only defective in inducing the release of IL-1 β but not TNF- α from M1 macrophages *in vitro*. Therefore, the defective in the NLRP3 inflammasome activation is likely responsible for the reduced production of inflammatory cytokines in mice infected by the *als* mutant. Therefore, this manuscript focuses on inflammasome activation with purified Als3.

Line 83:

Figure 1a The unpublished data cannot serve as reference. Besides, it is confusing that the authors use *als3* single deletion strain or triple mutant strain irregularly. If the authors try to demonstrate the biological function of Als protein family, experiments about other single deletion or mutant strains are needed. If the authors focus on the Als3-induced immune response, compare the difference between the Als protein family and select the most reasonable object to clarify in the context.

The unpublished data have been deleted in this revised version.

We focus on Als3-induced immune response, as the *als3* single mutant showed a more significant defect. In the revision, we compared the difference between different *als* mutants. As shown in new Fig. 1d, Fig. 2b, and supplementary Fig. 2b, *als* triple mutant is more defective than *als3* single mutant. The triple mutant was generated much later because of some technical limitations. So, we used the triple in all subsequent experiments. Triple, double or single *als* mutants all showed defects, differing only in severity.

To conclude “Live elongating hyphae induce higher level of IL-1beta compared to yeast”, the hyphae-locked strain such as *nrg1* or *tup1* are needed for comparison. Besides, the schematic diagram is not needed since it cannot provide valuable information.

Additional hyphal mutants are shown in supplemental figure 1a in the revised version. The schematic diagram is deleted.

Figure 1c and Supplementary Figure 1a

References or experimental evidences are required to illustrate the relationship between macrophage death and IL-1beta production.

IL-1 β release and cell death are not necessarily correlated (PMID: 27419363). To avoid getting involved in this controversial topic, we decided to only describe results. As stated in the results, only the *als3* mutant caused both reduced macrophage death and IL-1beta production.

Fig. 1d and Supplementary Figure 1c

It is not a consensus that the complemented strain can regard as the control, the WT strain SN250 is still required.

Both WT and the complemented strain are shown in new Fig. 1d. No difference has been observed so far between the two strains in our various *in vivo* and *in vitro* assays. The complemented strain has been generated through CRISPR/Cas9 technology by replacing both alleles of the *als3* deletion with the *ALS3* coding region, flanked by *ALS3* 5' and 3' untranslated regions (UTR).

Figure 1e

The CFU in kidney and brain is not enough to support the conclusion that “ALS3 is required for IL-1beta production and fungal clearance”. Does recombinant mIL-1beta can rescue the phenotype induced by *als3* deletion strain? The authors must detect the IL-1beta production, T

cell response and the activation of innate immune cells between *als3* deletion and ALS3 complemented strain infected mice.

The *als* triple mutant did show reduced immune responses including the production of multiple cytokines and chemokines during infection, while the *als3* single mutant is only significantly defective in TNF production. These data are in new Fig. 2b of the revision. IL-17 was also measured but undetectable at day 2 post infection, which is too early for IL-17 production. Due to funding limitation, we are unable to carry out further *in vivo* experiments. This will be a topic of future research.

Figure 2a

The authors' data show that the Als3 level is closely correlative with beta glucan. Does *als3* influence the exposure of beta glucan in the surface of *C. albicans*? Whether *als3* deletion strain induced decreased macrophage response shown in figure 1 is due to decreased beta glucan in fungi? This question also exists in Figure 3, 4 and 5. Purified hyphae Als3 are needed in all *in vitro* experiments.

This part has been changed. Hyphal-associated β -glucan exposure is not affected in the *als* mutant. This has been included in the text (supplementary Fig. 3b). β -glucan exposure is increased during germ-tube formation (PMID: 34995333).

Als3 purified from hyphae is included in almost every *in vitro* experiment in this version, including new Fig. 5 d,f, Fig. 6d, and complementary Fig. 6b.

Figure 2b&c

The experimental design lacks rigorous control and the annotation makes audience confused. As the authors pronounced that the most important factor that inducing *als3* expression in the fungal culture medium shown in figure 2b is GlcNAc, why did the authors compare the RPMI and SC in figure 3c? Besides, the annotation in the top of figure 2b would likely to tell audience the RPMI is GlcNAc free, which make it difficult to understand what the authors wish to convey.

Thanks for these helpful questions and suggestions. Sorry for the confusion. We rephrased this paragraph to de-emphasize the basal media and emphasize the confirmed results of GlcNAc-inducing Als3 expressions. Since there is no difference between basal media, we used the data from the SC medium and RPMI-glucose-free medium as basal media in Fig. b and c. We changed the annotation to de-emphasize the basal media and avoid misunderstanding.

Figure 3a

The authors use CD18 deficient BMDM to describe Als3 induced inflammasome activation is CR3 dependent. However, triple mutant strain induced IL-1 β and caspase-1 activation still significantly lower than complemented strain in CD18 deficient BMDM, which suggests that Als3 induced inflammasome activation may be CR3 independent.

CR3 only partially contributes to Als3-induced inflammasome activation as we described in the text. Als3 has a CR3-independent activity, as described in the discussion.

Figure 3c

The switch between BMDM and BMDC is confusing since it is not well established that BMDC cannot engulf fungus.

In this manuscript, BMDCs have been used to analyze purified Als3 protein because IL-1 β from BMDMs was undetectable at the Als3 protein concentrations used (15 $\mu\text{g ml}^{-1}$). Based on the

publication (PMID: 25063877), BMDCs tend to produce higher cytokine in shorter periods compared to BMDMs. BMDCs were chosen. Besides, BMDCs have been used to analyze *C. albicans*-induced inflammasome activation (PMID: 20401456). For these reasons, BMDCs were used in these experiments.

Figure 3d

It is not convincing to say that CR3 promotes als-mediated inflammasome activation, since the termination of p-Syk cannot serve as the evidence of activation of CR3 signaling pathway. More work about the Als3-CR3 axis must be provided. Besides, we do not encourage researchers to compare bands in different membranes.

Thank you for pointing this out. We have changed to “CR3 is involved in Als-mediated inflammasome activation” in section title and Figure 4 title. The last paragraph has been deleted, and some sentences have been modified to avoid misunderstanding.

All Western data/bands shown in this manuscript are from the same membranes. Dashed lines are to divide WT and mutants of BMDMs or *Candida*.

Figure 4

The biochemical method the author used is not solid enough to prove the author’s statement in abstract that “CR3 I domain recognize Als family proteins”. More in vitro binding assay include ELISA and immunofluorescence using purified Als and CR3 mutant should be performed to clarify the synergize effect between Als3 and beta-glucan. Previous studies can provide effective guidance (PMID: 37872182).

We have tried to create a foundational ELISA assay for CR3 binding in this short period.

Unfortunately, we tried several antibodies and couldn’t make the positive control working. So, we designed a novel particulate β -glucan pull-down assay. Our results are consistent with Als3 binding to the I-domain and synergy between Als3 and beta-glucan. The new results are shown in Fig. 5 d,f.

“we performed a pull-down with particulate β -glucan as the solid phase and soluble β -glucan or β -methylglucoside as antagonists. Als3 greatly increased CR3 binding to particulate β -glucan (Fig. 5d). Soluble β -glucan and β -methylglucoside both exhibited a partial blockage of the Als3-mediated interaction between CR3 and particulate β -glucan (Fig. 5d). In BMDMs, we also found that hyphal Als-mediated inflammasome activation was largely blocked by β -methylglucoside and soluble β -glucan (Fig. 5e), indicating that hyphal β -glucan is involved in the Als-mediated inflammasome signaling.

“In the particulate β -glucan pull-down assay, the Als-mediated interaction of CR3 and β -glucan was also significantly inhibited by the I-domain specific antibody ICRF44, but not its isotype control (Fig. 5f), further supporting an association of Als3 with the CD11b I domain and the synergy between the I domain and the lectin-like domain in ligand binding.”

Figure 5a

The *C. albicans* induced syk phosphorylation in CD18 deficient BMDMs are totally different with authors shown in figure 3d. This is paradoxical even if macrophage polarization may influence.

The images are from non-polarized M0 macrophages and the results match with Fig. a, b, c in this Figure. This doesn’t match with Syk phosphorylation with M1 macrophages in the previous

figure, because GM-CSF priming makes p-Syk very strong for both WT and *CD18* deficient BMDMs with no difference.

Figure 5d

It is hardly to observe the decreased clustering of p-Tyr and F-actin, nor the difference of F-actin cuff formation between *CD18* deficient and WT BMDMs. The fluorescence background seems not normalized neither. The data shown in figure 5d barely match the statement “Als promoting F-actin remodeling” in abstract. More significant images and quantification of protein co-localization are needed.

To set a proper excitation value within the maximum for the majority of pictures, a relatively weak laser energy was used, which might result in slightly unclear pictures. To address this, we have now adjusted the contrast consistently for pictures with the same fluorescence. Also, quantification is shown in supplementary Fig. 6c.

Figure 5e&f

The direct connection between *Als3* induced p-syk and inflammasome activation are not well illustrated. The Src family kinase downstream of beta-integrin play an important role in F-actin polymerization and signaling transduction, authors should test whether Src is involved in *Als3* induced inflammasome signaling. In addition, as the authors mentioned F-actin remodeling, it is most likely to influence the fungal phagocytosis, which has a great effort in “inside-out signaling” and immune response. The authors should provide that whether the macrophage phagocytosis is different to *als3* delete and control strain.

Yes, we agree. We think that they are two independently events controlled by *Als*-CR3. In GM-CSF-primed macrophages, the control strain and the *als* triple mutant induced similar levels of p-Syk (fig. 4d) and inflammasome-independent cytokine TNF- α (Fig. 4e), but the *als* mutant released a lower level of inflammasome-dependent cytokine IL-1 β than the control *C. albicans* strain. Therefore, *Als* proteins have an additional function for inflammasome activation in addition to its function in Syk activation (observed in M0 macrophages).

Since the Src family kinase is also downstream of CR3, we decided not to pursue further investigation into the Src family kinase pathway.

We did measure the phagocytosis of the control strains and the triple mutant in supplementary Fig. 4c and showed that the control strain was engulfed at 2 hours post-infection, which matches with the early termination of p-Syk.

Reviewer #3:

This paper examines a previously uncharacterized interaction between ALS3 and host molecules involved in the immune response and I think the work will be of interest to the field.

Comments

General

There are several references to precursor information that clearly led to this work that will be included in another manuscript in preparation for publication. It's unfortunate that submission of this paper didn't wait for those results to be published.

We deleted all sentences about the unpublished reference.

Overall the authors assume that the reader is familiar with the different sizes of inactive and active forms of the proteins examined by western blot. A little more information would make the results clearer to the non-immunologist.

The detailed description has been included in the figure legend of Fig. 1 d,e,f. and the main text.

There are several places where the descriptions are more casual than might be expected in a paper (like a notable surge). It would be better to use more formal and more accurate descriptions.

We used more accurate descriptions in this revised version.

The rationale for why particular mutant strains were used in particular experiments isn't always clear and there is inconsistency between using specific strain names vs. wild-type throughout the text and the figure labels.

We haven't found a significant difference between the *als3* single mutant and double mutants. So, we didn't differentiate them very strictly while designing the follow-up experiments. In this version, we have consistently used WT instead of SN250 or SC5314 in the figures to enhance the readability.

Results

Figure 1. The *flo8* mutant mentioned in figure 1a is not in the strain list. I think the 6 well plate cartoon could be omitted as it doesn't illustrate anything meaningful. Panels b and c have WT and SN250, respectively. Was the same strain used in both? What is the significance of the dashed line in panels b and c and why is it in a different location in each? The legend mentions unstimulated macrophages but the graphs are labelled uninfected.

This version has included the *flo8* mutant in the strain list.

The 6-well plate cartoon has been deleted. Panels b and c have only WT in this version. The right side of the dashed line shows control strains, not the mutants for screening. This has been explained in the text. Both the legend and the figures use "uninfected" in this version.

Lines 84/85. If the result wasn't significantly different, just say that instead of emphasizing differences.

The description has been changed according to this suggestion.

Line 94. It may be helpful to readers to describe these as low dose infections since they are substantially below what is used in a typical tail vein injection when testing virulence.

"Low dose" has been described in this revised version.

Figure 2. Why is a different wild-type strain being used here than in Figure 1 or Figure 3?

In revised Fig. 1d, both strains have been tested and there is no big difference between them.

Considering that the complemented strain *als3^{Δ/Δ}::ALS3* is also a CRISPR-Cas9 edited product as the mutant, it has been chosen as a control.

Line 102-105. Is this trying to say that GlcNAc was chosen because of a known effect on beta glucan exposure and that was a condition of interest? As written it seems to imply that GlcNAc in particular induces hyphae while other media do not, which isn't accurate, and it doesn't make clear the choice of GlcNAc over other media that also induce the expression of hypha specific genes.

This paragraph has been changed to: "Quantitative RT-PCR data confirmed that the hyphal expression of *ALS3*, but not *ALS1* and *ALS5*, was consistently significantly higher in the medium with GlcNAc than the medium without GlcNAc..."

Line 112. The result is significant or not. It can't be more significant than something else.

The sentence has been changed.

Lines 115-117. This description is confusing and seems more like discussion than results. Stating that Als proteins from hyphae are doing something makes it sound like Als proteins were isolated from hyphae and then used in the experiment.

This paragraph has been reorganized, and this description has been deleted.

Line 121. The text refers to wild-type, but Fig 3a is labelled Control, which elsewhere refers to a complement strain. What strain was used?

Thanks for pointing out this mistake. The "wild-type" in the text has been changed into "the control strain".

Figure 4. Why does panel b use Mn but panel c uses Mg?

With macrophages, we found manganese alone could activate NLRP3 inflammasome signaling (data not shown). Therefore, manganese wasn't used with cells. Manganese alone also has been reported to activate NLRP3 inflammasome signaling in microglial cells (PMID: 30622196).

On the other hand, previous binding experiments found that either manganese or magnesium could increase the I domain binding, but manganese-mediated binding is much stronger (PMID: 8458080; PMID: 7524101). Thus, in the binding experiment, we chose manganese.

Lines 176-178. "Moreover, EDTA treatment diminished the priming signaling induced by hyphae, as evidenced by the low pro-IL-1 β level, and the inflammasome signaling induced by Als proteins (Fig. 4c)." The part of this is clear in the figure. What is the second part of the statement referring to?

The sentence has been modified. The second part refers to caspase-1 p20.

Line 186/187. This sort of statement seems a bit out of place in a results section and certainly shouldn't be a single sentence paragraph.

The sentence has been deleted. Thanks.

Lines 199/204. The levels of p-Syk are highest in SN250, but the pattern is the same in all three strains (upper panel). The last sentence of the paragraph seems redundant.

The last sentence of the paragraph has been removed.

Line 221. Are the methodological details like reference to beads necessary?

We think the reference for the CBRM1/5 antibody is necessary, but the latter part of this sentence has been changed.

Line 230. It might be clearer to describe the results, not the conclusion.

Changed into: Our results showed that CytoD treatment weakened the difference between the control strain and the *als1^{SY}als3^{SY}als5^{Δ/Δ}* mutant but did not reduce the level in activating inflammasome. Lat B treatment reduced the levels of inflammasome signaling in both the control strain and the *als1^{SY}als3^{SY}als5^{Δ/Δ}* (Fig. 6g).

Discussion

Lines 263-272.

The importance of the comparison being made here isn't clear. Induction in one medium in vitro looked much like another, and all were similar to in vivo induction. The cited work examined cells grown in shaking broth cultures so the comparison to the work presented here with cells grown in dishes isn't direct.

This paragraph has been deleted.

The comments about the link to cancer feel tacked on and even as a speculation reaching too far. These sentences have been deleted.

Language

Many sentences that begin with an adverb, like notably in line 201, which could be omitted.

Author response: All of them have been omitted. Thanks.

Throughout, growing cells in broth in a well of a multi-well dish cannot be described as growth on a plate, which would instead refer to on a solid medium.

Changed, thanks.

Line 51 might be better as “Pra1, whose production is higher in hyphae”

Changed, thanks.

Line 122. Reduced or lower would be better than diminished. Restructuring the sentence might make the whole statement clearer.

Changed into “lower”.

Line 123/124. This statement needs a verb.

Changed. Thanks!

Line 153. Fix inline citation.

Removed. Thanks!

Line 159/160. “Our results demonstrated that” is unnecessary.

Removed. Thanks!

Line 189. Cut the first sentence of the paragraph.

Removed. Thanks!

Line 163/164. Rephrase to remove the redundancy of analyzing interaction by investigating interactions.

Rephrased to: “His-tagged Als3 and recombinant CR3, including the CD11b and CD18 subunits, were used to analyze protein-protein interactions.”.

Line 212-217. Rephrase to remove redundancy.

Rechecked. It seems there is no redundancy in these sentences. The former sentences are about “inside-out signaling” and the latter are about “outside-in” signaling.

Line 225 should read phosphatases.

Changed. Thanks!

Line 241/242 should read “recognize” and “promote”.

Changed. Thanks!

Line 257-260. The statement starting Furthermore is a run-on sentence and needs to be reworked.

Removed “Furthermore” and deleted the last sentence of this paragraph. Thanks!

Line 284-286. Please rephrase these awkward sentences.

Rephrased to: “The interactions among PRRs play a crucial role in the innate immune response against microbes⁶⁷. CR3 collaborates with multiple receptors, such as TLR2 and Dectin-1, contributing to the elimination of microbes from the host^{28, 57, 68}.”.

REVIEWER COMMENTS

Reviewer #1 (Remarks to the Author):

In general, the revisions have answered my concerns. I have three suggestions for further clarification of figures.

Fig. 1e. Relabel the "control" lane as "control (*als3Δ/Δ::ALS3*).

Fig. 5a relabel the second lane as *als3Δ/Δ*

Suppl. Fig 5 and 6d: addition of longer exposure images would help to make the data more compelling.

Reviewer #2 (Remarks to the Author):

In this revised manuscript, the authors have suitably addressed my main concern and questions. I recommend that this revised manuscript could be accepted for publication in Nature Communications after minor revisions.

Minor points:

1. They should perform quantitative analysis of all WB bands.
2. The transcription level of IL-1beta need be examined to rule out the possibility that mutant strains affect Syk-mediated IL-1beta expression.
3. The survival and fungal burden of NLRP3- and CD18-knockout mice need be examined to conclude that CR3-dependent NLRP3 activation is involved in the promotion of fungal clearance by Als family protein.

Reviewer #3 (Remarks to the Author):

In the strain table in Supplementary Fig 1 the description should be defective, not defected.

RESPONSE TO REVIEWER COMMENTS

Reviewer #1 (Remarks to the Author):

In general, the revisions have answered my concerns. I have three suggestions for further clarification of figures.

Fig. 1e. Relabel the "control" lane as "control (*als3Δ/Δ::ALS3*).

Changed, thanks.

Fig. 5a relabel the second lane as *als3Δ/Δ*

Relabeled, thanks.

Suppl. Fig 5 and 6d: addition of longer exposure images would help to make the data more compelling.

Suppl Fig 5 images have been replaced by longer exposure images.

The original Suppl Fig 6d (current Suppl Fig 6f) is already a long exposure image. Longer exposure makes the background too high.

Reviewer #2 (Remarks to the Author):

In this revised manuscript, the authors have suitably addressed my main concern and questions. I recommend that this revised manuscript could be accepted for publication in Nature Communications after minor revisions.

Minor points:

1. They should perform quantitative analysis of all WB bands.

In this version, WB quantification has been integrated into almost all WB figures, including Figures 3c, 4a, 5a, 5c, 5e, 6g, and Figures 6a, 6c (in Suppl Fig.6a,6c). WB quantification is not included in Fig. 1, because it mainly showed IL-1 β level that has been quantified by ELISA. Caspase-1 p20 level of the strains were quantified in other figures (Fig.3c, Fig. 4a, Fig. 5c, 5e).

2. The transcription level of IL-1beta need be examined to rule out the possibility that mutant strains affect Syk-mediated IL-1beta expression.

RNA levels are irrelevant given that pro-IL-1 β protein are not affected by Als in WBs.

3. The survival and fungal burden of NLRP3- and CD18-knockout mice need be examined to conclude that CR3-dependent NLRP3 activation is involved in the promotion of fungal clearance by Als family protein.

We provided mechanistic evidence of Als-CR3 interaction through biochemistry and signaling assays. It's important to emphasize that while Als-CR3 interaction contributes to Als3-mediated NLRP3 inflammasome activation, CR3 is not the sole receptor for Als proteins. Therefore, the expectation is not that *als* mutants would behave exactly like the wild-type strain in CD18-knockout mice. It's well established from previous publications that both host NLRP3 and CD18 are crucial for optimal clearance of *C. albicans*. In this manuscript, the role of hyphal Als proteins as ligands in *C. albicans* clearance is clearly demonstrated.

Reviewer #3 (Remarks to the Author):

In the strain table in Supplementary Fig 1 the description should be defective, not defected.

Changed, thanks.